# Hepatitis B, C and D virus infections and risk of hepatocellular carcinoma in Africa: A meta-analysis including sensitivity analyses for studies comparable for confounders

**Donatien Serge Mbaga**[1]*, **Sebastien Kenmoe**[2]*, **Cyprien Kengne-Ndé**[3], **Jean Thierry Ebogo-Belobo**[4], **Gadji Mahamat**[1], **Joseph Rodrigue Foe-Essomba**[5], **Marie Amougou-Atsama**[6], **Serges Tchatchouang**[7], **Inès Nyebe**[1], **Alfloditte Flore Feudjio**[8], **Ginette Irma Kame-Ngasse**[4], **Jeannette Nina Magoudjou-Pekam**[8], **Lorraine K. M. Fokou**[8], **Dowbiss Meta-Djomsi**[6], **Martin Maïdadi-Foudi**[6], **Sabine Aimee Touangnou-Chamda**[1], **Audrey Gaelle Daha-Tchoffo**[9], **Abdel Aziz Selly-Ngaloumo**[8], **Rachel Audrey Nayang-Mundo**[10], **Jacqueline Félicité Yéngué**[11], **Jean Bosco Taya-Fokou**[1], **Raoul Kenfack-Momo**[8], **Efietngab Atembeh Noura**[4], **Cynthia Paola Demeni Emoh**[1], **Hervé Raoul Tazokong**[1], **Arnol Bowo-Ngandji**[1], **Carole Stéphanie Sake**[1], **Etienne Atenguena Okobalemba**[12], **Jacky Njiki Bikoi**[1], **Richard Njouom**[2], **Sara Honorine Riwom Essama**[1]

**1** Department of Microbiology, The University of Yaounde I, Yaoundé, Cameroon, **2** Virology Department, Centre Pasteur of Cameroon, Yaoundé, Cameroon, **3** Evaluation and Research Unit, National AIDS Control Committee, Yaoundé, Cameroon, **4** Medical Research Centre, Institute of Medical Research and Medicinal Plants Studies, Yaoundé, Cameroon, **5** Department of Mycobacteriology, Centre Pasteur of Cameroon, Yaoundé, Cameroon, **6** Centre de Recherche sur les Maladies Émergentes et Re-Emergentes, Institute of Medical Research and Medicinal Plants Studies, Yaoundé, Cameroon, **7** Bacteriology Department, Centre Pasteur of Cameroon, Yaoundé, Cameroon, **8** Department of Biochemistry, The University of Yaounde I, Yaoundé, Cameroon, **9** Department of Medical Biochemistry, The University of Yaounde I, Yaoundé, Cameroon, **10** Department of Microbiology, Protestant University of Central Africa, Yaoundé, Cameroon, **11** Department of Animals Biology and Physiology, The University of Yaounde I, Yaoundé, Cameroon, **12** Faculty of Medicine and Biomedical Science, The University of Yaoundé I, Yaoundé, Cameroon

* mbaga2015.mds@gmail.com (DSM); sebastien.kenmoe@ubuea.cm (SK)

## Abstract

### Introduction

Africa denotes unique facies for hepatocellular carcinoma (HCC) characterized by a conjunction of low sensitization, restricted access to diagnosis and treatment and associated with the highest incidence and mortality in the world. We investigated whether hepatitis B (HBV), C (HCV) and D (VHD) viruses were etiological agents of HCC in Africa.

### Methods

Relevant articles were searched in PubMed, Web of Science, African Index Medicus, and African Journal Online databases, as well as manual searches in relevant reviews and included articles. Analytical studies from Africa evaluating the association between HCC development and HBV, HCV, and HDV were included. Relevant studies were selected, data extracted, and the risk of bias assessed independently by at least 2 investigators. The association was estimated using odds ratios (OR) and their 95% confidence interval (95% CI)

**Data Availability Statement:** All relevant data are within the paper and its Supporting Information files.

**Funding:** This project is part of the EDCTP2 programme supported by the European Union under grant agreement TMA2019PF-2705. "The funders had no role in study design, data collection and analysis, decision to publish, or preparation of the manuscript."

**Competing interests:** The authors have declared that no competing interests exist.

determined by a random-effects model. Sources of heterogeneity were determined by sub-group analyses.

## Results

A total of 36 case-control studies were included. With controls having non-hepatic disease, the overall results suggested a significantly increased risk of HCC in patients with HBV (HBeAg (OR = 19.9; 95% CI = [3.7–105.2]), HBsAg (OR = 9.9; 95%) CI = [6.2–15.6]) and DNA (OR = 8.9; 95% CI = [5.9–13.4]); HCV (Anti-HCV (OR = 9.4; 95% CI = [6.3–14.0]) and RNA (OR = 16.5; 95% CI = [7.8–34.6]); HDV (Anti-VHD, (OR = 25.8; 95% CI = [5.9–112.2]); and HBV/HCV coinfections (HBV DNA/HCV RNA (OR = 22.5; 95% CI = [1.3–387.8]). With apparently healthy controls, the overall results suggested a significantly increased risk of HCC in patients with HBV (HBsAg, (OR = 8.9; 95% CI = [6.0–13.0]); HCV (Anti-HCV, (OR = 7.7; 95% CI = [5.6–10.6]); and HBV/HCV coinfections (HBsAg/Anti-HCV (OR = 7.8; 95% CI = [4.4–13.6]) Substantial heterogeneity and the absence of publication bias were recorded for these results.

## Conclusions

In Africa, HBV/HCV coinfections and HBV, HCV, and HDV infections are associated with an increased risk of developing HCC. The implementation of large-scale longitudinal and prospective studies including healthy participants to search for early biomarkers of the risk of progression to HCC is urgently needed.

## Introduction

Hepatocellular carcinoma (HCC) is one of the most common cancer and the third leading cause of death due to malignancy worldwide [1]. HCC incidence is continuing increasing with more than 900 thousand new cases and as many deaths recorded worldwide in 2020 [2]. HCC rates vary widely across the world with the greatest burden reported in Southeast Asia, East Asia, and sub-Saharan Africa [1]. Almost 80% of the morbidity and mortality due to HCC is attributed to developing countries [3].

Hepatitis B Virus (HBV) and Hepatitis C Virus (HCV) are the main etiologic agents of HCC [4, 5]. More than half of the HCC cases are attributable to HBV while about 25% are attributable to HCV. Controversial results have been reported on the increased risk of HBV/HCV coinfections compared to HBV or HCV monoinfections [6–9]. The role of Hepatitis D Virus (HDV) and occult hepatitis B in the development of HCC has also been demonstrated [10, 11]. The etiological factors of HCC vary considerably across regions of the world and indicate a strong disparity in the distribution of incidences of HCC [12]. The areas of high HBV prevalence are also those with the highest rates of HCC. Africa, which is a region with a high endemicity for HBV, thus has the highest incidences of HCC with more than 15 incident cases per 100 thousand inhabitants [13–16]. Key reviews, which included very few African studies, reported the importance of genotyping and mutations of interleukin-6 and HBV as biomarkers for early identification of the risk of progression to HCC [17–19]. Africa is a unique region plagued by limited awareness, late diagnosis, and limited access to care and treatment for HCC [20]. The critical burden of HCC in Africa is mostly related to late diagnosis and limited access to treatment [21–23]. A precise synthesis on the roles of the main HCC viral etiological factors

(HBV, HCV, and HDV) would be important to enlighten health decision makers on preventive and early diagnosis methods specific for the African region.

We conducted a systematic review to determine whether HBV, HCV, HDV infections and HBV/HCV coinfections are associated with an increased risk of developing HCC in Africa.

## Methods

### Literature search

Electronic literature search was performed for studies published from databases inception through February 2020 and updated in March 2021. Searches were conducted in Pubmed, Web of Science, African Index Medicus, and African Journal Online. The strategy included the keyword combination for exposure (HBV, HCV and HDV), outcome (HCC) and context (Africa) (S1 Table). Additional potentially relevant studies were searched manually from the reference list of included articles and relevant reviews. The protocol for this review was declared in the international PROSPERO database (CRD42020181381) and complied with the PRISMA guidelines (S2 Table).

### Inclusion and exclusion criteria

Regardless of antiviral therapy status, comparative studies (clinical trials, cohort, and case control) examining the relationship between infection with HBV, HCV, HDV, and HBV/HCV coinfections and the risk of developing HCC were considered relevant for this review. Only studies in French or English conducted in Africa were considered. We considered all types of HCC diagnosed by clinical, histological, biochemical, and radiological approaches. The controls were apparently healthy people, people with non-hepatic diseases, people with liver cirrhosis, and people with hepatic diseases other than liver cirrhosis (other liver diseases). All the detection techniques for HBV (HBsAg, HBeAg and HBV DNA), HCV (anti-HCV antibodies and HCV RNA) and HDV (anti-HDV antibodies, AgHDV and HDV RNA) infection markers were considered. Studies excluded were those without control groups, with participant selection bias, with no full text and/or abstracts available, conducted outside of Africa, duplicates, case reports, and reviews.

### Selection, data extraction and risk of bias assessment of included studies

Studies were selected on the basis of a title/abstract screening according to the study inclusion criteria. Full texts of selected studies were reviewed to validate eligibility and data extracted from included studies. Data retrieved were name of first author, year of publication, study design, sampling approach, timing of testing for hepatitis virus infection (retrospective/prospective), country, UNSD region, country income level, period of recruitment of study participants, study context (rural/urban and community/hospital). We also collected inclusion criteria of participants, the definition of HCC, the inclusion criteria for controls, socio-demographic confounding factors, other non-viral confounding factors known to be associated with the risk of HCC, biochemical parameters of liver, type of hepatic virus (HBV, HCV or HDV), the hepatitis virus detection method, the marker searched for the detection of hepatitis virus, the number of participants and controls and the number of exposed and unexposed subjects. Data for the assessment of individual risk of bias of studies using the Newcastle-Ottawa method (S3 Table). Discussion and consensus were used to resolve issues in the event of disagreements between investigators.

## Statistical analysis

The parameters used to estimate the association between viral hepatitis and the risk of developing hepatocellular carcinoma were odds ratios (OR) and the corresponding to 95% confidence intervals determined by random effect meta-analysis [24]. Quantitative analysis was performed using the packages "meta" and "metafor" of R software version 4.0.3 software [25, 26]. Heterogeneity was assessed for each analysis using Cochrane's Q-test and $I^2$ measurement [27]. P value $\leq 0.10$ or $I^2 \geq 50\%$ indicates significant heterogeneity. The publication bias was examined by the Egger test and funnel plot, and P value $\leq 0.10$ indicate the presence of a publication bias [28]. Sensitivity analysis was performed to evaluate the validity and reliability of overall results on studies with low risk of bias and those comparable for confounding factors [29].

## Results

### Study selection

The article selection process is presented in Fig 1. A total of 5469 articles were identified from our electronic literature search and 21 manually from the reviewed bibliography. We excluded 1299 duplicates, 4076 based on title and abstract screening, and 79 for various reasons including wrong study population or design and unavailable article full texts (S4 Table). We thus had a total of 36 studies (114 effect data) that fit the inclusion criteria for this review [30–65].

### Included studies characteristics

Bias assessment scores ranged from 3 to 9 and the median was 8 [IQR = 6–8]. Out of the 114-effect data, 75 (65.7%) were of low risk of bias, 37 (32.4%) of moderate risk of bias and 2 (1.7) of high risk of bias. The risk of bias of the effect data is shown in S5 Table. Participants in the included studies were recruited between 1981 and 2015 and studies published between 1975 and 2020 (S6 Table). All effect data included in this meta-analysis had a case-control design and the majority had non-probability sampling (105/114). No study was representative of a national population. Collection of data on hepatitis status was prospective in half of the effect data (56/114). Most of the effect data came from West Africa (52/114) and North Africa (27/114). Most of the included effects data were from The Gambia (19/114), Egypt (18/114) and South Africa (14/114). Half of the effect data came from low-middle income countries (57/114). None of the included study authors stated that the study was conducted in a rural area and most were in a hospital setting (89/114). Controls in effect data were predominantly people with non-hepatic disease, mostly matched for age and gender. Sixty-two effect data were for HBV, 35 for HCV, 12 for HDV and 5 for HBV/HCV coinfection. The majority of the effect data used multiple HCC diagnostic approach including clinical, biochemical, radiological and histological methods (79/114). More than half of the effect data had confirmed HCC histologically (70/114). The most widely used hepatitis virus detection techniques were radioimmunoassay (26/114), indirect (24/114) and direct (19/114) Enzyme Linked Immunosorbent Assay, and Enzyme immunoassay (16/114). The HBV (HBeAg, HBsAg and DNA), HCV (Anti-HCV and RNA) and HDV (HDV Ag, Anti HDV and RNA) markers studies used 4 types of controls including apparently healthy controls, controls with non-hepatic diseases, controls with cirrhosis of the liver and controls with other liver diseases. The most represented markers were HBsAg (48/114), Anti-HCV (33/114) and Ag Delta (8/114) for HBV, HCV and HDV respectively. The primary characteristics of the endpoint data are presented in S7 Table.

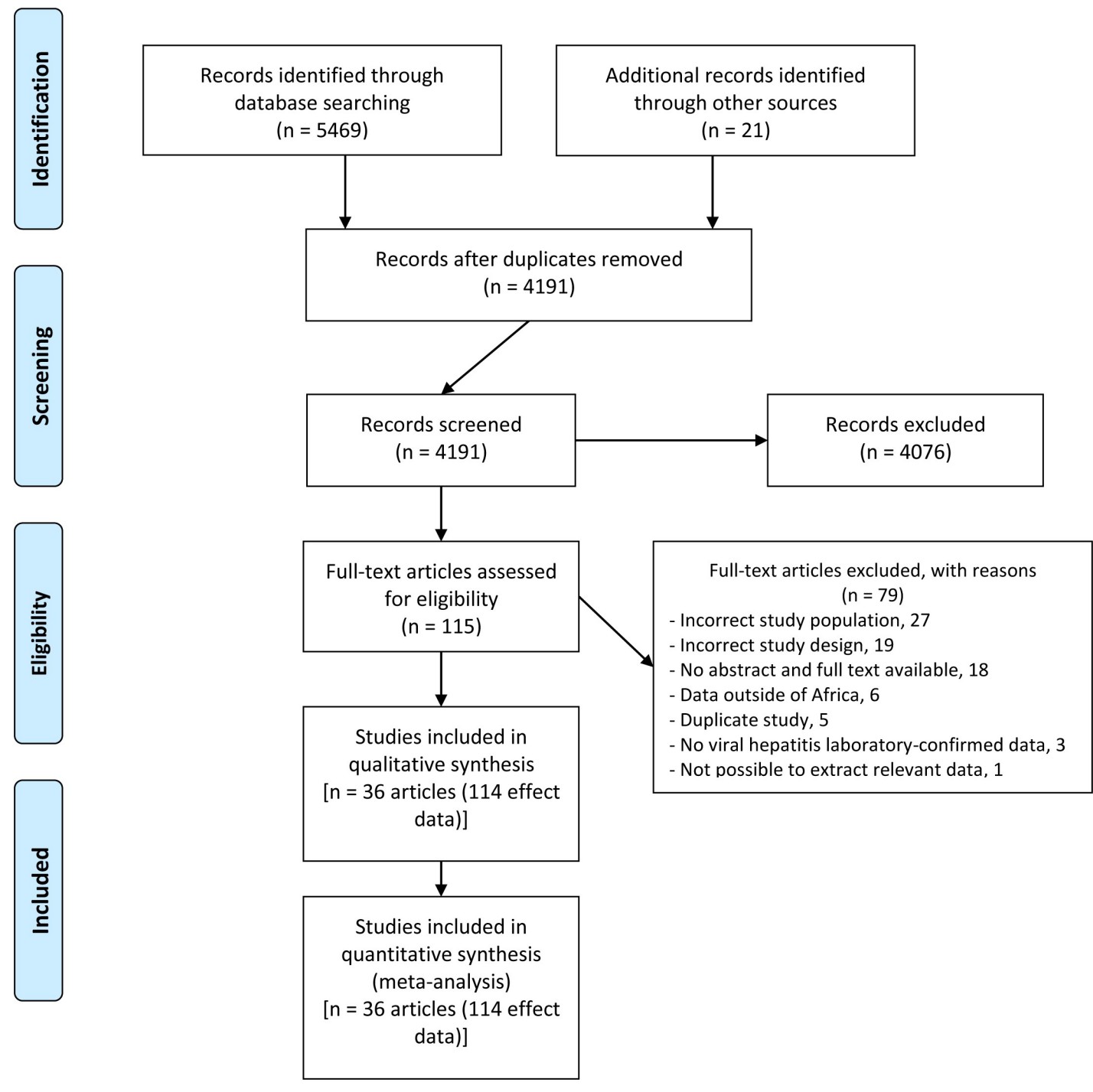

**Fig 1. PRISMA flow-chart of studies selected for the meta-analysis.**

## Meta-analysis Hepatitis B Virus

HBeAg (OR = 19.9; 95% CI = [3.7–105.2]), HBsAg (OR = 9.9; 95% CI = [6.2–15.6]) and DNA (OR = 8.9; 95% CI = [5.9–13.4]) for HBV were associated with an increased risk of developing HCC with controls with non-hepatic disease (Fig 2). HBsAg was further associated with an

| Study or Subgroup | HCC (+) Events | Total | HCC (−) Events | Total | Weight | OR [95% CI] |
|---|---|---|---|---|---|---|
| **HBV_HBeAg Positive_Healthy controls** | | | | | | |
| Coursaget, 1978 | 1 | 103 | 1 | 100 | 0.9% | 0.97 [ 0.06; 15.73] |
| Cronberg, 1984 | 6 | 175 | 3 | 50 | 1.5% | 0.56 [ 0.13; 2.31] |
| Kirk, 2005 | 21 | 186 | 2 | 348 | 1.5% | 22.02 [ 5.10; 95.02] |
| Ola, 2012 | 5 | 41 | 0 | 45 | 0.8% | 13.71 [ 0.73; 256.19] |
| **Random effect meta-analysis** | | 505 | | 543 | 4.7% | 3.49 [ 0.62; 19.61] |
| Heterogeneity: Tau² = 1.9631; Chi² = 14.14, df = 3 (P = 0.0027); I² = 78.8% [43.1%; 92.1%] | | | | | | |
| **HBV_HBeAg Positive_Liver cirrhosis** | | | | | | |
| Coursaget, 1978 | 1 | 103 | 6 | 76 | 1.1% | 0.11 [ 0.01; 0.97] |
| Kirk, 2005 | 21 | 186 | 14 | 94 | 1.9% | 0.73 [ 0.35; 1.50] |
| **Random effect meta-analysis** | | 289 | | 170 | 3.0% | 0.60 [ 0.30; 1.20] |
| Heterogeneity: Tau² = < 0.0001; Chi² = 2.58, df = 1 (P = 0.1085); I² = 61.2% [ 0.0%; 91.0%] | | | | | | |
| **HBV_HBeAg Positive_Non-hepatic diseases** | | | | | | |
| Dhifallah, 2020 | 6 | 73 | 0 | 70 | 0.8% | 13.58 [ 0.75; 245.72] |
| Tswana, 1992 | 20 | 102 | 1 | 100 | 1.2% | 24.15 [ 3.17; 183.77] |
| **Random effect meta-analysis** | | 175 | | 170 | 2.0% | 19.98 [ 3.79; 105.27] |
| Heterogeneity: Tau² = 0; Chi² = 0.1, df = 1 (P = 0.7497); I² = 0% | | | | | | |
| **HBV_HBeAg Positive_Other liver disorders** | | | | | | |
| Cronberg, 1984 | 6 | 175 | 4 | 83 | 1.6% | 0.70 [ 0.19; 2.56] |
| Marchio, 2018 | 31 | 195 | 39 | 263 | 1.9% | 1.09 [ 0.65; 1.81] |
| **Random effect meta-analysis** | | 370 | | 346 | 3.5% | 1.02 [ 0.64; 1.65] |
| Heterogeneity: Tau² = 0; Chi² = 0.38, df = 1 (P = 0.5379); I² = 0% | | | | | | |
| **HBV_HBsAg Positive_Healthy controls** | | | | | | |
| Brown, 1984 | 21 | 36 | 0 | 70 | 0.8% | 195.58 [11.23; 3405.78] |
| Chin'ombe, 2009 | 29 | 60 | 4 | 30 | 1.6% | 6.08 [ 1.89; 19.55] |
| Coursaget, 1978 | 70 | 103 | 12 | 100 | 1.8% | 15.56 [ 7.49; 32.33] |
| Coursaget, 1992 | 22 | 49 | 52 | 134 | 1.9% | 1.28 [ 0.66; 2.49] |
| Cronberg, 1984 | 119 | 175 | 14 | 50 | 1.9% | 5.46 [ 2.73; 10.94] |
| Hassan, 2001 | 5 | 33 | 1 | 35 | 1.1% | 6.07 [ 0.67; 55.04] |
| Kew, 1979 | 178 | 289 | 24 | 213 | 1.9% | 12.63 [ 7.76; 20.54] |
| Kew, 1990 | 184 | 380 | 9 | 152 | 1.9% | 14.92 [ 7.39; 30.13] |
| Kirk, 2005 | 88 | 186 | 47 | 348 | 2.0% | 5.75 [ 3.77; 8.76] |
| Larouzé, 1977 | 10 | 21 | 2 | 38 | 1.4% | 16.36 [ 3.11; 86.20] |
| Mahale, 2019 | 250 | 431 | 80 | 470 | 2.0% | 6.73 [ 4.95; 9.16] |
| Mboto, 2005 | 5 | 13 | 5 | 39 | 1.5% | 4.25 [ 0.99; 18.29] |
| Mets, 1993 | 12 | 26 | 5 | 54 | 1.6% | 8.40 [ 2.53; 27.90] |
| Montaser, 2007 | 15 | 32 | 0 | 10 | 0.8% | 18.60 [ 1.00; 344.26] |
| Nishioka, 1975 | 23 | 42 | 21 | 450 | 1.8% | 24.73 [11.69; 52.30] |
| Ola, 2012 | 26 | 41 | 0 | 45 | 0.8% | 155.58 [ 8.94; 2707.44] |
| Omer, 2001 | 49 | 115 | 14 | 199 | 1.9% | 9.81 [ 5.09; 18.93] |
| Soliman, 2010 | 132 | 148 | 74 | 150 | 1.9% | 8.47 [ 4.61; 15.59] |
| **Random effect meta-analysis** | | 2180 | | 2587 | 28.7% | 8.91 [ 6.08; 13.05] |
| Heterogeneity: Tau² = 0.4039; Chi² = 63.67, df = 17 (P < 0.0001); I² = 73.3% [57.4%; 83.3%] | | | | | | |
| **HBV_HBsAg Positive_Liver cirrhosis** | | | | | | |
| Cenac, 1987 | 21 | 29 | 38 | 55 | 1.7% | 1.17 [ 0.43; 3.18] |
| Coursaget, 1978 | 70 | 103 | 62 | 76 | 1.9% | 0.48 [ 0.23; 0.98] |
| Coursaget, 1992 | 22 | 49 | 14 | 48 | 1.8% | 1.98 [ 0.85; 4.58] |
| Kirk, 2005 | 88 | 186 | 40 | 94 | 1.9% | 1.21 [ 0.74; 2.00] |
| Mahale, 2019 | 183 | 312 | 67 | 119 | 2.0% | 1.10 [ 0.72; 1.69] |
| Mets, 1993 | 12 | 26 | 13 | 79 | 1.7% | 4.35 [ 1.64; 11.52] |
| Nishioka, 1975 | 23 | 42 | 13 | 18 | 1.6% | 0.47 [ 0.14; 1.54] |
| **Random effect meta-analysis** | | 747 | | 489 | 12.6% | 1.17 [ 0.74; 1.84] |
| Heterogeneity: Tau² = 0.2210; Chi² = 16.85, df = 6 (P = 0.0098); I² = 64.4% [19.7%; 84.2%] | | | | | | |
| **HBV_HBsAg Positive_Non-hepatic diseases** | | | | | | |
| Amr, 2014 | 85 | 132 | 190 | 669 | 2.0% | 4.56 [ 3.07; 6.76] |
| Bahri, 2011 | 29 | 164 | 9 | 250 | 1.8% | 5.75 [ 2.64; 12.51] |
| Cenac, 1987 | 21 | 29 | 1 | 46 | 1.1% | 118.13 [13.86; 1006.42] |
| Dhifallah, 2020 | 17 | 73 | 6 | 70 | 1.7% | 3.24 [ 1.19; 8.78] |
| Ezzat, 2005 | 18 | 236 | 8 | 236 | 1.8% | 2.35 [ 1.00; 5.52] |
| Jaquet, 2018 | 34 | 40 | 7 | 80 | 1.6% | 59.10 [18.45; 189.24] |
| Jaquet, 2018 | 30 | 44 | 7 | 88 | 1.7% | 24.80 [ 9.13; 67.36] |
| Jaquet, 2018 | 48 | 76 | 17 | 152 | 1.9% | 13.61 [ 6.85; 27.05] |
| Kew, 1986 | 54 | 124 | 2 | 62 | 1.5% | 23.14 [ 5.41; 98.94] |
| Larouzé, 1977 | 31 | 39 | 6 | 53 | 1.6% | 30.35 [ 9.60; 96.01] |
| Lightfoot, 1997 | 160 | 167 | 148 | 167 | 1.8% | 2.93 [ 1.20; 7.18] |
| Mandishona, 1998 | 16 | 24 | 5 | 48 | 1.6% | 17.20 [ 4.90; 60.40] |
| Marchio, 2018 | 115 | 195 | 5 | 49 | 1.7% | 12.65 [ 4.81; 33.30] |
| Mohamed, 1992 | 33 | 101 | 5 | 101 | 1.7% | 9.32 [ 3.46; 25.09] |
| Olubuyide, 1997 | 38 | 64 | 32 | 64 | 1.9% | 1.46 [ 0.73; 2.94] |
| Skelton, 2000 | 74 | 148 | 11 | 148 | 1.9% | 12.45 [ 6.22; 24.92] |
| Tabor, 1977 | 12 | 19 | 3 | 40 | 1.5% | 21.14 [ 4.71; 94.86] |
| Tabor, 1977 | 22 | 47 | 3 | 50 | 1.6% | 13.79 [ 3.76; 50.60] |
| Tswana, 1992 | 102 | 182 | 11 | 100 | 1.9% | 10.32 [ 5.17; 20.60] |
| **Random effect meta-analysis** | | 1904 | | 2473 | 32.2% | 9.91 [ 6.28; 15.63] |
| Heterogeneity: Tau² = 0.7564; Chi² = 84.92, df = 18 (P < 0.0001); I² = 78.8% [67.5%; 86.2%] | | | | | | |
| **HBV_HBsAg Positive_Other liver disorders** | | | | | | |
| Cenac, 1987 | 21 | 29 | 12 | 28 | 1.7% | 3.50 [ 1.16; 10.58] |
| Cronberg, 1984 | 119 | 175 | 45 | 83 | 1.9% | 1.79 [ 1.05; 3.07] |
| Marchio, 2018 | 115 | 195 | 164 | 263 | 2.0% | 0.87 [ 0.59; 1.27] |
| Montaser, 2007 | 15 | 32 | 5 | 15 | 1.6% | 1.76 [ 0.49; 6.34] |
| **Random effect meta-analysis** | | 431 | | 389 | 7.1% | 1.48 [ 0.89; 2.45] |
| Heterogeneity: Tau² = 0.1291; Chi² = 8.9, df = 3 (P = 0.0306); I² = 66.3% [ 1.3%; 88.5%] | | | | | | |
| **HBV_DNA_Liver cirrhosis** | | | | | | |
| Gouas, 2012 | 33 | 198 | 15 | 78 | 1.9% | 0.84 [ 0.43; 1.65] |
| **Random effect meta-analysis** | | 198 | | 78 | 1.9% | 0.84 [ 0.43; 1.65] |
| Heterogeneity: not applicable | | | | | | |
| **HBV_DNA_Non-hepatic diseases** | | | | | | |
| Gouas, 2012 | 33 | 198 | 2 | 325 | 1.5% | 32.30 [ 7.66; 136.27] |
| Mak, 2018 | 80 | 150 | 56 | 438 | 2.0% | 7.80 [ 5.09; 11.94] |
| Schiefelbein, 2012 | 15 | 148 | 0 | 114 | 0.9% | 26.59 [ 1.57; 449.29] |
| **Random effect meta-analysis** | | 496 | | 877 | 4.3% | 8.94 [ 5.97; 13.40] |
| Heterogeneity: Tau² = 0; Chi² = 4.03, df = 2 (P = 0.1336); I² = 50.3% [ 0.0%; 85.6%] | | | | | | |

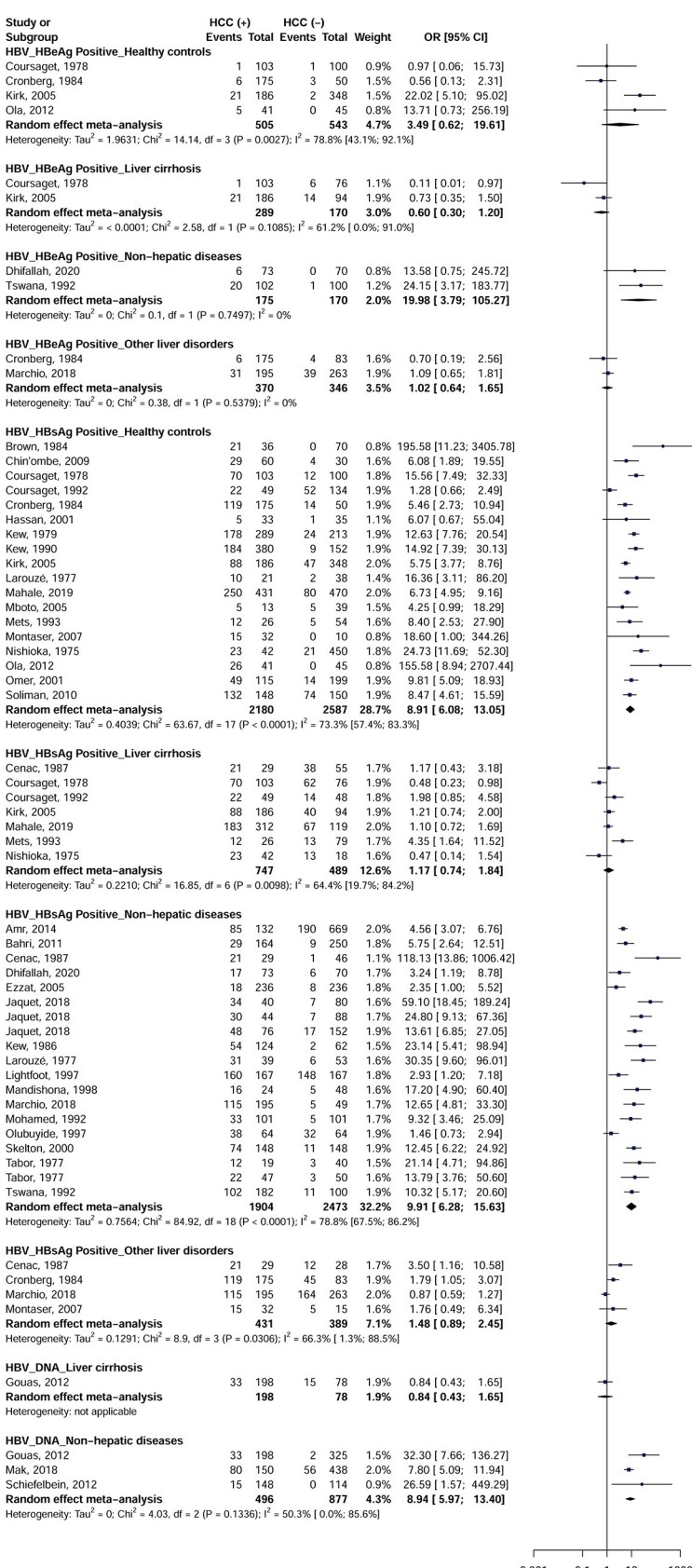

0.001 0.1 1 10 1000

**Fig 2. Association between Hepatitis B Virus infection and risk of hepatocellular carcinoma in Africa.**

increased risk of developing HCC in apparently healthy controls (OR = 8.9; 95% CI = [6.0–13.0]). No significant difference was observed for HBeAg with apparently healthy controls (OR = 3.4; 95% CI = [0.6–19.6]), controls with liver cirrhosis (OR = 0.6; 95% CI = [0.3–1.2]), and controls with other liver diseases (OR = 1.0; 95% CI = [0.6–1.6]). No significant difference was observed for HBsAg with controls with liver cirrhosis (OR = 1.1; 95% CI = [0.7–1.8]) and controls with other liver diseases (OR = 1.4; 95% CI = [0.8–2.4]). No significant difference was observed for HBV DNA in a study of control with liver cirrhosis (OR = 0.8; 95% CI = [0.4–1.6]). For categories with 3 or more effect data, heterogeneity was significant (I2> 50%) for all analyses according to the types of HBV marker and controls.

### Meta-analysis Hepatitis C Virus

Anti-HCV (OR = 9.4; 95% CI = [6.3–14.0]) and RNA (OR = 16.5; 95% CI = [7.8–34.6]) of HCV were associated with increased risk of developing HCC with controls with non-hepatic disease (Fig 3). Anti-HCV was also associated with an increased risk of developing HCC in apparently healthy controls (OR = 7.7; 95% CI = [5.6–10.6]). There were no studies of HCV RNA with apparently healthy individuals, controls with cirrhosis of the liver, and those with other liver diseases. No significant difference was observed for anti-HCV with controls with liver cirrhosis (OR = 1.9; 95% CI = [0.9–3.8]) and controls with other liver diseases (OR = 1.1; 95% CI = [0.7–1.7]). For categories with 3 or more outcome data, with the exception of anti-HCV with apparently healthy controls, heterogeneity was significant ($I^2> 50\%$) for the rest of the analyzes by type of marker HCV and controls.

### Meta-analysis Hepatitis B Virus/Hepatitis C Virus co-infections

HBV DNA/HCV RNA (OR = 22.5; 95% CI = [1.3–387.8]) of HBV/HCV coinfection were associated with an increased risk of developing HCC with controls with non-hepatic disease (Fig 4). HBsAg/anti-HCV (OR = 7.8; 95% CI = [4.4–13.6]) of HBV/HCV coinfection were associated with an increased risk of developing HCC with apparently healthy controls. No significant difference was observed for HBsAg/anti-HCV with controls with non-hepatic diseases (OR = 1.8; 95% CI = [0.5–6.6]). Only one category of analyzes on HBV/HCV coinfection had 3 or more effect data and showed no heterogeneity in the estimation of the association between HBsAg/anti-HCV and HCC with apparently healthy controls.

### Meta-analysis Hepatitis D Virus

No positive HDV antigen was found in two studies and no meta-analysis was possible. Most of the categories analyzed had less than 3 studies and it is therefore difficult to draw conclusions on the results obtained. Anti-HDV was associated with a risk of developing HCC with controls with non-hepatic disease (OR = 25.8; 95% CI = [5.9–112.2], 3 studies) (Fig 5). Only one category of HDV assays had 3 or more effect data and showed no heterogeneity in the estimation of the association between anti-HDV and HCC with controls with non-hepatic disease.

### Publication bias and sensitivity analysis

Egger's linear regression test found no evidence of significant publication bias for any analysis category with at least 3 effect data (p> 0.005; Table 1). This result was confirmed by funnel plots which showed no asymmetry (S1–S11 Figs). We performed a sensitivity analysis for categories with 3 or more outcome data by selecting only studies with low risk of bias and studies comparable for confounding factors identified in the included studies. S8 and S9 Tables present the distribution between cases and controls for covariates known to be associated with the

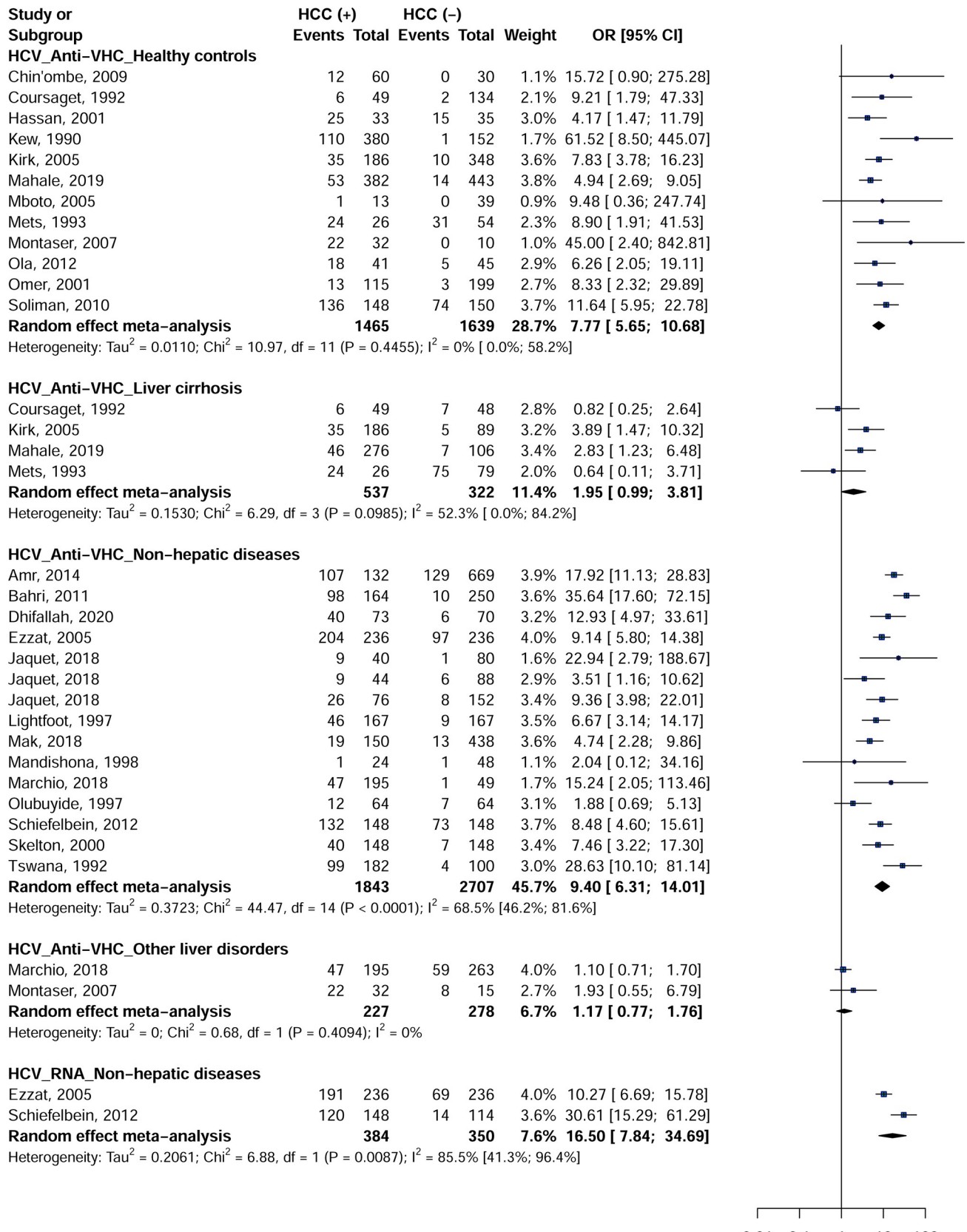

| Study or Subgroup | HCC (+) Events | Total | HCC (−) Events | Total | Weight | OR [95% CI] |
|---|---|---|---|---|---|---|
| **HCV_Anti−VHC_Healthy controls** | | | | | | |
| Chin'ombe, 2009 | 12 | 60 | 0 | 30 | 1.1% | 15.72 [ 0.90; 275.28] |
| Coursaget, 1992 | 6 | 49 | 2 | 134 | 2.1% | 9.21 [ 1.79; 47.33] |
| Hassan, 2001 | 25 | 33 | 15 | 35 | 3.0% | 4.17 [ 1.47; 11.79] |
| Kew, 1990 | 110 | 380 | 1 | 152 | 1.7% | 61.52 [ 8.50; 445.07] |
| Kirk, 2005 | 35 | 186 | 10 | 348 | 3.6% | 7.83 [ 3.78; 16.23] |
| Mahale, 2019 | 53 | 382 | 14 | 443 | 3.8% | 4.94 [ 2.69; 9.05] |
| Mboto, 2005 | 1 | 13 | 0 | 39 | 0.9% | 9.48 [ 0.36; 247.74] |
| Mets, 1993 | 24 | 26 | 31 | 54 | 2.3% | 8.90 [ 1.91; 41.53] |
| Montaser, 2007 | 22 | 32 | 0 | 10 | 1.0% | 45.00 [ 2.40; 842.81] |
| Ola, 2012 | 18 | 41 | 5 | 45 | 2.9% | 6.26 [ 2.05; 19.11] |
| Omer, 2001 | 13 | 115 | 3 | 199 | 2.7% | 8.33 [ 2.32; 29.89] |
| Soliman, 2010 | 136 | 148 | 74 | 150 | 3.7% | 11.64 [ 5.95; 22.78] |
| **Random effect meta−analysis** | | **1465** | | **1639** | **28.7%** | **7.77 [ 5.65; 10.68]** |

Heterogeneity: Tau$^2$ = 0.0110; Chi$^2$ = 10.97, df = 11 (P = 0.4455); I$^2$ = 0% [ 0.0%; 58.2%]

| Study or Subgroup | HCC (+) Events | Total | HCC (−) Events | Total | Weight | OR [95% CI] |
|---|---|---|---|---|---|---|
| **HCV_Anti−VHC_Liver cirrhosis** | | | | | | |
| Coursaget, 1992 | 6 | 49 | 7 | 48 | 2.8% | 0.82 [ 0.25; 2.64] |
| Kirk, 2005 | 35 | 186 | 5 | 89 | 3.2% | 3.89 [ 1.47; 10.32] |
| Mahale, 2019 | 46 | 276 | 7 | 106 | 3.4% | 2.83 [ 1.23; 6.48] |
| Mets, 1993 | 24 | 26 | 75 | 79 | 2.0% | 0.64 [ 0.11; 3.71] |
| **Random effect meta−analysis** | | **537** | | **322** | **11.4%** | **1.95 [ 0.99; 3.81]** |

Heterogeneity: Tau$^2$ = 0.1530; Chi$^2$ = 6.29, df = 3 (P = 0.0985); I$^2$ = 52.3% [ 0.0%; 84.2%]

| Study or Subgroup | HCC (+) Events | Total | HCC (−) Events | Total | Weight | OR [95% CI] |
|---|---|---|---|---|---|---|
| **HCV_Anti−VHC_Non−hepatic diseases** | | | | | | |
| Amr, 2014 | 107 | 132 | 129 | 669 | 3.9% | 17.92 [11.13; 28.83] |
| Bahri, 2011 | 98 | 164 | 10 | 250 | 3.6% | 35.64 [17.60; 72.15] |
| Dhifallah, 2020 | 40 | 73 | 6 | 70 | 3.2% | 12.93 [ 4.97; 33.61] |
| Ezzat, 2005 | 204 | 236 | 97 | 236 | 4.0% | 9.14 [ 5.80; 14.38] |
| Jaquet, 2018 | 9 | 40 | 1 | 80 | 1.6% | 22.94 [ 2.79; 188.67] |
| Jaquet, 2018 | 9 | 44 | 6 | 88 | 2.9% | 3.51 [ 1.16; 10.62] |
| Jaquet, 2018 | 26 | 76 | 8 | 152 | 3.4% | 9.36 [ 3.98; 22.01] |
| Lightfoot, 1997 | 46 | 167 | 9 | 167 | 3.5% | 6.67 [ 3.14; 14.17] |
| Mak, 2018 | 19 | 150 | 13 | 438 | 3.6% | 4.74 [ 2.28; 9.86] |
| Mandishona, 1998 | 1 | 24 | 1 | 48 | 1.1% | 2.04 [ 0.12; 34.16] |
| Marchio, 2018 | 47 | 195 | 1 | 49 | 1.7% | 15.24 [ 2.05; 113.46] |
| Olubuyide, 1997 | 12 | 64 | 7 | 64 | 3.1% | 1.88 [ 0.69; 5.13] |
| Schiefelbein, 2012 | 132 | 148 | 73 | 148 | 3.7% | 8.48 [ 4.60; 15.61] |
| Skelton, 2000 | 40 | 148 | 7 | 148 | 3.4% | 7.46 [ 3.22; 17.30] |
| Tswana, 1992 | 99 | 182 | 4 | 100 | 3.0% | 28.63 [10.10; 81.14] |
| **Random effect meta−analysis** | | **1843** | | **2707** | **45.7%** | **9.40 [ 6.31; 14.01]** |

Heterogeneity: Tau$^2$ = 0.3723; Chi$^2$ = 44.47, df = 14 (P < 0.0001); I$^2$ = 68.5% [46.2%; 81.6%]

| Study or Subgroup | HCC (+) Events | Total | HCC (−) Events | Total | Weight | OR [95% CI] |
|---|---|---|---|---|---|---|
| **HCV_Anti−VHC_Other liver disorders** | | | | | | |
| Marchio, 2018 | 47 | 195 | 59 | 263 | 4.0% | 1.10 [ 0.71; 1.70] |
| Montaser, 2007 | 22 | 32 | 8 | 15 | 2.7% | 1.93 [ 0.55; 6.79] |
| **Random effect meta−analysis** | | **227** | | **278** | **6.7%** | **1.17 [ 0.77; 1.76]** |

Heterogeneity: Tau$^2$ = 0; Chi$^2$ = 0.68, df = 1 (P = 0.4094); I$^2$ = 0%

| Study or Subgroup | HCC (+) Events | Total | HCC (−) Events | Total | Weight | OR [95% CI] |
|---|---|---|---|---|---|---|
| **HCV_RNA_Non−hepatic diseases** | | | | | | |
| Ezzat, 2005 | 191 | 236 | 69 | 236 | 4.0% | 10.27 [ 6.69; 15.78] |
| Schiefelbein, 2012 | 120 | 148 | 14 | 114 | 3.6% | 30.61 [15.29; 61.29] |
| **Random effect meta−analysis** | | **384** | | **350** | **7.6%** | **16.50 [ 7.84; 34.69]** |

Heterogeneity: Tau$^2$ = 0.2061; Chi$^2$ = 6.88, df = 1 (P = 0.0087); I$^2$ = 85.5% [41.3%; 96.4%]

0.01  0.1  1  10  100

**Fig 3. Association between Hepatitis C Virus infection and risk of hepatocellular carcinoma in Africa.**

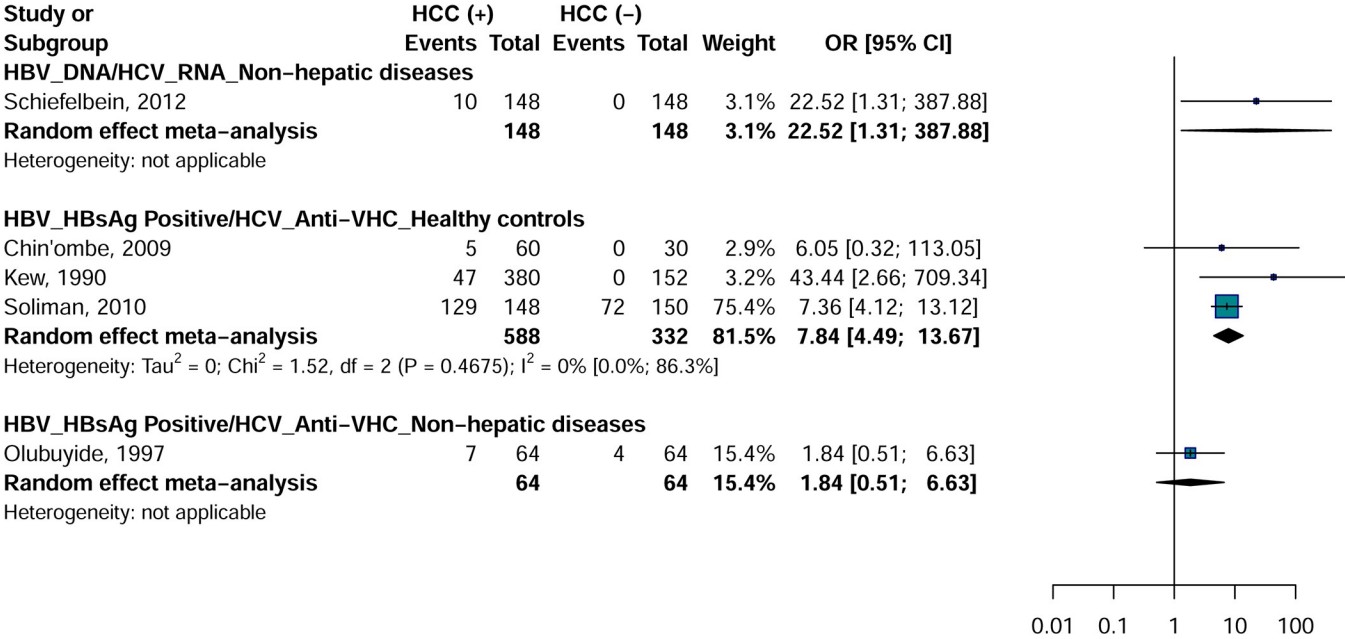

**Fig 4. Association between Hepatitis B and C Virus coinfection and risk of hepatocellular carcinoma in Africa.**

risk of HCC. The selection of these studies for sensitivity analysis did not change the overall trends (Table 1). A few exceptions were recorded for certain confounding factors represented by a single study.

## Source of heterogeneity examination

In the additional sub-analysis, a study with data from HBeAg infections collected retrospectively (p = 0.001), conducted in The Gambia (p = 0.001), in a low-resource setting (p = 0.001), after the years 2000 (p <0.001) using enzyme immunoassays for HBeAg detections (p = 0.001) had a significantly higher effect of the association between HBeAg and HCC with apparently healthy controls (OR = 22.0; 95% CI = [5.1–95.0]) (S10 Table) [46]. The association between HBsAg and HCC with apparently healthy controls was maintained in all categories of further sub-analysis. This effect, however, varied statistically significantly depending on the country (p = 0.005) and the method of hepatitis detection (p = 0.030). An increase of the effect size between HBsAg and HCC with apparently healthy controls depending on the country's income level (p = 0.021) was observed. No difference was observed in the subgroup analysis of the effect between HBsAg and HCC with controls with liver cirrhosis. With controls having non-hepatic diseases, the effect between HBsAg and HCC varied significantly by country (p <0.001), UNSD region (effect highest in West Africa, p = 0.001), level of country income (effect higher in low-income countries, p = 0.032) and hepatitis detection technique (effect higher with rapid diagnostic tests, p <0.001). With controls with other hepatic diseases, the effect between HBsAg and HCC varied significantly depending on the timing of data collection on hepatitis virus infection (high effect in prospective studies, p = 0.005), the UNSD region (size of the high effect in West Africa, p = 0.021) and the year of publication (size of the high effect in articles published before 2000, p = 0.021). No difference was observed in the effect analysis subgroups between HBV DNA and HCC with controls with non-hepatic disease. No difference was observed in the subgroup analysis of the effect between anti-HCV and HCC with apparently healthy controls. With controls having liver cirrhosis, the effect between anti-

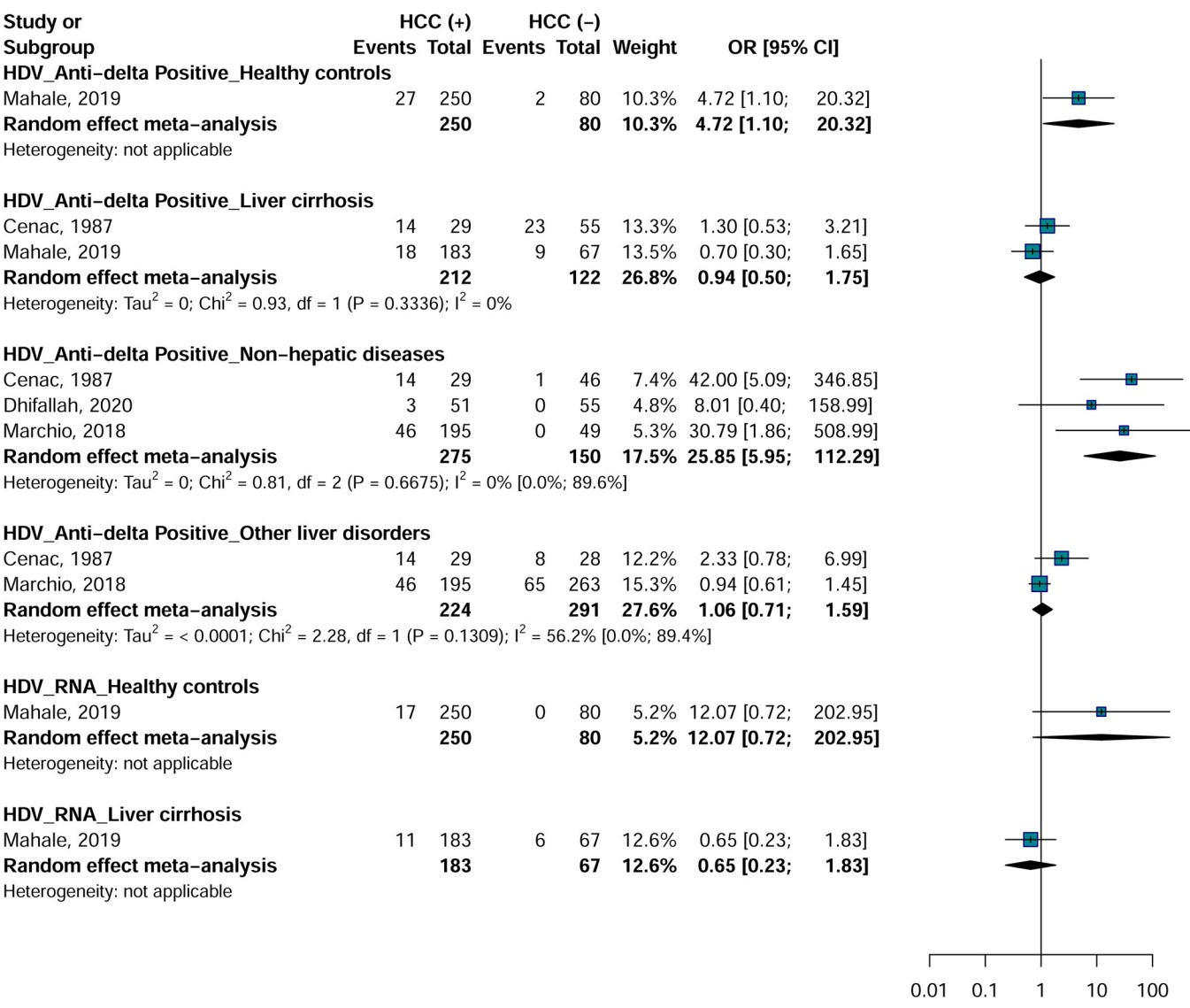

**Fig 5. Association between Hepatitis D Virus infection and risk of hepatocellular carcinoma in Africa.**

HCV and HCC varied significantly depending on the timing of data collection for hepatitis virus infection (high effect size for retrospective studies, p = 0.014), country (highest effect size in Gambia, p = 0.049) and publication year (high effect size in articles published after 2000, p = 0.014). With controls having non-hepatic disease, the effect between anti-HCV and HCC varied significantly by country (p = 0.002) and UNSD region (highest effect size in East Africa, p = 0.006). No significant difference was observed in the analysis subgroups of the effect between anti-HDV and HCC with the controls with non-hepatic disease and HBV/HCV and HCC coinfection with the apparently healthy controls.

## Discussion

This review shows a quantitative overview of the relative contribution of HBV, HCV and HDV in the development of HCC in 20 countries of the 5 African USND regions. As expected, HBV, HCV and HDV markers were important contributors to the development of HCC with

**Table 1. Hepatocellular carcinoma development in people with and without viral hepatitis B, C, and D infections in Africa and influence of confounders.**

| | OR (95%CI) | 95% Prediction interval | N Studies | N HCC cases | N controls | H (95% CI) | I² (95%CI) | P heterogeneity | P Egger test |
|---|---|---|---|---|---|---|---|---|---|
| **HBV** | | | | | | | | | |
| **HBeAg** | | | | | | | | | |
| **Healthy controls** | | | | | | | | | |
| Overall | 3.5 [0.6–19.6] | [0–4317] | 4 | 505 | 543 | 2.2 [1.3–3.6] | 78.8 [43.1–92.1] | 0.003 | 0.329 |
| Risk of bias | 3.5 [0.6–19.6] | [0–4317] | 4 | 505 | 543 | 2.2 [1.3–3.6] | 78.8 [43.1–92.1] | 0.003 | 0.329 |
| Male gender | 3.4 [0.5–25.7] | NA | 2 | 144 | 145 | 1.3 | 39.4 | 0.199 | NA |
| **HBsAg** | | | | | | | | | |
| **Healthy controls** | | | | | | | | | |
| Overall | 8.9 [6.1–13.1] | [2.2–36.4] | 18 | 2180 | 2587 | 1.9 [1.5–2.4] | 73.3 [57.4–83.3] | 0 | 0.214 |
| Risk of bias | 8.9 [6.9–11.5] | [4.7–16.7] | 12 | 1934 | 1814 | 1.4 [1–1.9] | 46.3 [0–72.5] | 0.039 | 0.142 |
| Alcohol drinking | 7.2 [5.5–9.5] | [1.2–43.1] | 3 | 579 | 704 | 1 [1–2.3] | 0 [0–80.3] | 0.59 | 0.834 |
| Anti-Schistosoma IgG | 6.1 [0.7–55] | NA | 1 | 33 | 35 | NA | NA | 1 | NA |
| Cancer family history | 8.5 [4.6–15.6] | NA | 1 | 148 | 150 | NA | NA | 1 | NA |
| Male gender | 11.2 [7.8–15.9] | [7.2–17.3] | 8 | 509 | 651 | 1.2 [1–1.9] | 35.5 [0–71.5] | 0.145 | 0.387 |
| Married | 7.1 [5.4–9.3] | NA | 2 | 579 | 620 | 1 | 0 | 0.509 | NA |
| Oral contraceptive user | 6.1 [0.7–55] | NA | 1 | 33 | 35 | NA | NA | 1 | NA |
| Smoke exposure | 8.3 [4.6–14.9] | NA | 2 | 181 | 185 | 1 | 0 | 0.775 | NA |
| Age | 9.2 [5.2–16.2] | NA | 2 | 169 | 188 | 1 | 0 | 0.466 | NA |
| **Liver cirrhosis** | | | | | | | | | |
| Overall | 1.2 [0.7–1.8] | [0.3–4.5] | 7 | 747 | 489 | 1.7 [1.1–2.5] | 64.4 [19.7–84.2] | 0.01 | 0.724 |
| Risk of bias | 1.1 [0.7–1.7] | NA | 1 | 312 | 119 | NA | NA | 1 | NA |
| Alcohol drinking | 1.1 [0.7–1.7] | NA | 1 | 312 | 119 | NA | NA | 1 | NA |
| Earth floor house | 1.1 [0.7–1.7] | NA | 1 | 312 | 119 | NA | NA | 1 | NA |
| Education (some) | 1.1 [0.7–1.7] | NA | 1 | 312 | 119 | NA | NA | 1 | NA |
| Family history of cancer | 1.1 [0.7–1.7] | NA | 1 | 312 | 119 | NA | NA | 1 | NA |
| Male gender | 1.4 [0.3–6.3] | NA | 2 | 129 | 155 | 3.6 [1.9–6.6] | 92.2 [73.4–97.7] | 0 | NA |
| Married | 1.1 [0.7–1.7] | NA | 1 | 312 | 119 | NA | NA | 1 | NA |
| Age | 4.4 [1.6–11.5] | NA | 1 | 26 | 79 | NA | NA | 1 | NA |
| **Non-hepatic diseases** | | | | | | | | | |
| Overall | 9.9 [6.3–15.6] | [1.5–66.2] | 19 | 1904 | 2473 | 2.2 [1.8–2.7] | 78.8 [67.5–86.2] | 0 | 0.237 |
| Risk of bias | 9.7 [6.2–15.4] | [1.6–60.5] | 17 | 1802 | 2357 | 2.2 [1.7–2.7] | 78.9 [66.9–86.6] | 0 | 0.282 |
| Alcohol drinking | 18.2 [11.2–29.4] | [6.2–53.4] | 5 | 285 | 469 | 1.3 [1–2.1] | 40.6 [0–78.1] | 0.151 | 0.577 |
| Born in rural areas | 2.4 [1–5.5] | NA | 1 | 236 | 236 | NA | NA | 1 | NA |
| Diabetes mellitus | 12.7 [4.8–33.3] | NA | 1 | 195 | 49 | NA | NA | 1 | NA |

*(Continued)*

**Table 1.** (Continued)

| | OR (95%CI) | 95% Prediction interval | N Studies | N HCC cases | N controls | H (95% CI) | I² (95%CI) | P heterogeneity | P Egger test |
|---|---|---|---|---|---|---|---|---|---|
| Diet cassava consumption | 12.7 [4.8–33.3] | NA | 1 | 195 | 49 | NA | NA | 1 | NA |
| Diet groundnut consumption | 12.7 [4.8–33.3] | NA | 1 | 195 | 49 | NA | NA | 1 | NA |
| Diet maize consumption | 12.7 [4.8–33.3] | NA | 1 | 195 | 49 | NA | NA | 1 | NA |
| Diet rice consumption | 12.7 [4.8–33.3] | NA | 1 | 195 | 49 | NA | NA | 1 | NA |
| Diet sorghum consumption | 12.7 [4.8–33.3] | NA | 1 | 195 | 49 | NA | NA | 1 | NA |
| Education (some) | 9.1 [3–27.7] | [0–5312239.1] | 3 | 356 | 476 | 2.7 [1.6–4.6] | 86.5 [61.2–95.3] | 0.001 | 0.777 |
| Ever helping in farming | 2.4 [1–5.5] | NA | 1 | 236 | 236 | NA | NA | 1 | NA |
| Ever working in farming | 2.4 [1–5.5] | NA | 1 | 236 | 236 | NA | NA | 1 | NA |
| HIV coinfection | 23 [12–44.2] | [0.1–9519.1] | 3 | 160 | 320 | 1.5 [1–2.9] | 57.2 [0–87.8] | 0.097 | 0.293 |
| Iron overload | 17.2 [4.9–60.4] | NA | 1 | 24 | 48 | NA | NA | 1 | NA |
| Male gender | 7.8 [4.2–14.5] | [0.9–69.2] | 12 | 1263 | 1196 | 2.1 [1.6–2.8] | 77.9 [61.7–87.2] | 0 | 0.217 |
| Occupation, None | 1.5 [0.7–2.9] | NA | 1 | 64 | 64 | NA | NA | 1 | NA |
| Other alcoholic beverages | 17.2 [4.9–60.4] | NA | 1 | 24 | 48 | NA | NA | 1 | NA |
| Pesticides at home | 2.4 [1–5.5] | NA | 1 | 236 | 236 | NA | NA | 1 | NA |
| Presence of HGV | 2.9 [1.2–7.2] | NA | 1 | 167 | 167 | NA | NA | 1 | NA |
| Previous blood transfusion | 3.9 [1–15.1] | NA | 2 | 246 | 164 | 3.9 [2.2–7] | 93.4 [78.5–98] | 0 | NA |
| Rodenticides at field | 2.4 [1–5.5] | NA | 1 | 236 | 236 | NA | NA | 1 | NA |
| Rodenticides at home | 2.4 [1–5.5] | NA | 1 | 236 | 236 | NA | NA | 1 | NA |
| Smoke exposure | 11.2 [4.7–26.7] | [0.5–249.2] | 5 | 648 | 618 | 2.3 [1.5–3.5] | 81 [55.5–91.8] | 0 | 0.849 |
| Surgical antecedents | 1.5 [0.7–2.9] | NA | 1 | 64 | 64 | NA | NA | 1 | NA |
| Tattoos-scarifications | 3.7 [1.4–9.6] | [0–303110.2] | 3 | 319 | 234 | 2.8 [1.6–4.7] | 86.9 [62.6–95.4] | 0 | 0.512 |
| Use of intravenous drug | 1.5 [0.7–2.9] | NA | 1 | 64 | 64 | NA | NA | 1 | NA |
| Aflatoxin B1-albumin adducts (pg_mg) | 17.2 [4.9–60.4] | NA | 1 | 24 | 48 | NA | NA | 1 | NA |
| Age | 6.7 [2.6–16.9] | [0.2–197.7] | 5 | 425 | 436 | 2.7 [1.8–4] | 86.5 [70.7–93.8] | 0 | 0.774 |
| Erythrocyte sedimentation rate (mm_hr) | 17.2 [4.9–60.4] | NA | 1 | 24 | 48 | NA | NA | 1 | NA |
| Hemoglobin (g_dL) | 17.2 [4.9–60.4] | NA | 1 | 24 | 48 | NA | NA | 1 | NA |
| Transferrin saturation (%) | 17.2 [4.9–60.4] | NA | 1 | 24 | 48 | NA | NA | 1 | NA |
| **Other liver disorders** | | | | | | | | | |
| Overall | 1.5 [0.9–2.5] | [0.2–9.9] | 4 | 431 | 389 | 1.7 [1–2.9] | 66.3 [1.3–88.5] | 0.031 | 0.227 |
| Risk of bias | 1.8 [1–3.1] | NA | 1 | 175 | 83 | NA | NA | 1 | NA |
| Diabetes mellitus | 0.9 [0.6–1.3] | NA | 1 | 195 | 263 | NA | NA | 1 | NA |
| Diet cassava consumption | 0.9 [0.6–1.3] | NA | 1 | 195 | 263 | NA | NA | 1 | NA |

*(Continued)*

**Table 1.** (Continued)

| | OR (95%CI) | 95% Prediction interval | N Studies | N HCC cases | N controls | H (95% CI) | I² (95%CI) | P heterogeneity | P Egger test |
|---|---|---|---|---|---|---|---|---|---|
| Diet maize consumption | 0.9 [0.6–1.3] | NA | 1 | 195 | 263 | NA | NA | 1 | NA |
| Diet rice consumption | 0.9 [0.6–1.3] | NA | 1 | 195 | 263 | NA | NA | 1 | NA |
| Diet sorghum consumption | 0.9 [0.6–1.3] | NA | 1 | 195 | 263 | NA | NA | 1 | NA |
| Liver cirrhosis | 1.8 [0.5–6.3] | NA | 1 | 32 | 15 | NA | NA | 1 | NA |
| Male gender | 2.6 [1.1–6] | NA | 2 | 61 | 43 | 1 | 0 | 0.427 | NA |
| Non-BDC | 0.9 [0.6–1.3] | NA | 1 | 195 | 263 | NA | NA | 1 | NA |
| Smoke exposure | 0.9 [0.6–1.3] | NA | 1 | 195 | 263 | NA | NA | 1 | NA |
| **DNA** | | | | | | | | | |
| **Non-hepatic diseases** | | | | | | | | | |
| Overall | 8.9 [6–13.4] | [0.7–123] | 3 | 496 | 877 | 1.4 [1–2.6] | 50.3 [0–85.6] | 0.134 | 0.802 |
| Risk of bias | 8.9 [6–13.4] | [0.7–123] | 3 | 496 | 877 | 1.4 [1–2.6] | 50.3 [0–85.6] | 0.134 | 0.802 |
| Alcohol drinking | 7.8 [5.1–11.9] | NA | 1 | 150 | 438 | NA | NA | 1 | NA |
| Diabetes mellitus | 7.8 [5.1–11.9] | NA | 1 | 150 | 438 | NA | NA | 1 | NA |
| HIV coinfection | 7.8 [5.1–11.9] | NA | 1 | 150 | 438 | NA | NA | 1 | NA |
| Male gender | 8 [5.3–12.2] | NA | 2 | 298 | 552 | 1 | 0 | 0.4 | NA |
| Married | 7.8 [5.1–11.9] | NA | 1 | 150 | 438 | NA | NA | 1 | NA |
| Residence in rural area | 8 [5.3–12.2] | NA | 2 | 298 | 552 | 1 | 0 | 0.4 | NA |
| Smoke exposure | 7.8 [5.1–11.9] | NA | 1 | 150 | 438 | NA | NA | 1 | NA |
| Age | 7.8 [5.1–11.9] | NA | 1 | 150 | 438 | NA | NA | 1 | NA |
| **HCV** | | | | | | | | | |
| **Anti-HCV** | | | | | | | | | |
| **Healthy controls** | | | | | | | | | |
| Overall | 7.8 [5.6–10.7] | [5–12] | 12 | 1465 | 1639 | 1 [1–1.5] | 0 [0–58.2] | 0.445 | 0.177 |
| Risk of bias | 8.2 [5.7–11.9] | [4.4–15.1] | 8 | 1297 | 1386 | 1.1 [1–1.7] | 24.4 [0–65.5] | 0.235 | 0.128 |
| Alcohol drinking | 5.1 [3.2–8.3] | [0.2–118.7] | 3 | 530 | 677 | 1 [1–1.9] | 0 [0–71.1] | 0.697 | 0.63 |
| Anti-Schistosoma IgG | 4.2 [1.5–11.8] | NA | 1 | 33 | 35 | NA | NA | 1 | NA |
| Cancer family history | 11.6 [5.9–22.8] | NA | 1 | 148 | 150 | NA | NA | 1 | NA |
| Male gender | 10 [6–16.7] | [4.4–23] | 5 | 349 | 443 | 1 [1–1.5] | 0 [0–57.7] | 0.742 | 0.317 |
| Married | 7.4 [4.1–13.4] | NA | 2 | 530 | 593 | 1.9 [1–3.9] | 71 [0–93.5] | 0.063 | NA |
| Oral contraceptive user | 4.2 [1.5–11.8] | NA | 1 | 33 | 35 | NA | NA | 1 | NA |
| Smoke exposure | 8.4 [4.5–15.6] | NA | 2 | 181 | 185 | 1.6 [1–3.4] | 62.2 [0–91.3] | 0.104 | NA |
| Age | 11.6 [5.9–22.8] | NA | 1 | 148 | 150 | NA | NA | 1 | NA |
| **Liver cirrhosis** | | | | | | | | | |

(Continued)

**Table 1.** (Continued)

| | OR (95%CI) | 95% Prediction interval | N Studies | N HCC cases | N controls | H (95% CI) | I² (95%CI) | P heterogeneity | P Egger test |
|---|---|---|---|---|---|---|---|---|---|
| Overall | 1.9 [1–3.8] | [0.2–18.2] | 4 | 537 | 322 | 1.4 [1–2.5] | 52.3 [0–84.2] | 0.099 | 0.112 |
| Risk of bias | 2.8 [1.2–6.5] | NA | 1 | 276 | 106 | NA | NA | 1 | NA |
| Alcohol drinking | 2.8 [1.2–6.5] | NA | 1 | 276 | 106 | NA | NA | 1 | NA |
| Earth floor house | 2.8 [1.2–6.5] | NA | 1 | 276 | 106 | NA | NA | 1 | NA |
| Education (some) | 2.8 [1.2–6.5] | NA | 1 | 276 | 106 | NA | NA | 1 | NA |
| Family history of cancer | 2.8 [1.2–6.5] | NA | 1 | 276 | 106 | NA | NA | 1 | NA |
| Male gender | 0.6 [0.1–3.7] | NA | 1 | 26 | 79 | NA | NA | 1 | NA |
| Married | 2.8 [1.2–6.5] | NA | 1 | 276 | 106 | NA | NA | 1 | NA |
| Age | 0.6 [0.1–3.7] | NA | 1 | 26 | 79 | NA | NA | 1 | NA |
| **Non-hepatic diseases** | | | | | | | | | |
| Overall | 9.4 [6.3–14] | [2.3–37.7] | 15 | 1843 | 2707 | 1.8 [1.4–2.3] | 68.5 [46.2–81.6] | 0 | 0.095 |
| Risk of bias | 9.2 [6–14] | [2.1–40] | 14 | 1770 | 2637 | 1.8 [1.4–2.4] | 70.6 [49.3–82.9] | 0 | 0.107 |
| Alcohol drinking | 5.9 [3.6–9.4] | [2.7–12.7] | 5 | 334 | 806 | 1.1 [1–2.3] | 10 [0–81.3] | 0.349 | 0.58 |
| Born in rural areas | 9.1 [5.8–14.4] | NA | 1 | 236 | 236 | NA | NA | 1 | NA |
| Diabetes mellitus | 5.4 [2.7–10.8] | NA | 2 | 345 | 487 | 1.1 | 12.9 | 0.284 | NA |
| Diet cassava consumption | 15.2 [2–113.5] | NA | 1 | 195 | 49 | NA | NA | 1 | NA |
| Diet groundnut consumption | 15.2 [2–113.5] | NA | 1 | 195 | 49 | NA | NA | 1 | NA |
| Diet maize consumption | 15.2 [2–113.5] | NA | 1 | 195 | 49 | NA | NA | 1 | NA |
| Diet rice consumption | 15.2 [2–113.5] | NA | 1 | 195 | 49 | NA | NA | 1 | NA |
| Diet sorghum consumption | 15.2 [2–113.5] | NA | 1 | 195 | 49 | NA | NA | 1 | NA |
| Education (some) | 8.2 [5.6–12] | [0.7–94.6] | 3 | 356 | 476 | 1.1 [1–3.5] | 22 [0–91.9] | 0.278 | 0.792 |
| Ever helping in farming | 9.1 [5.8–14.4] | NA | 1 | 236 | 236 | NA | NA | 1 | NA |
| Ever working in farming | 9.1 [5.8–14.4] | NA | 1 | 236 | 236 | NA | NA | 1 | NA |
| HIV coinfection | 6 [3.7–9.8] | [2.1–17.5] | 4 | 310 | 758 | 1.1 [1–2.9] | 22.8 [0–88.2] | 0.274 | 0.943 |
| Iron overload | 2 [0.1–34.2] | NA | 1 | 24 | 48 | NA | NA | 1 | NA |
| Male gender | 8.2 [4.8–14.1] | [1.5–44.9] | 9 | 1221 | 1470 | 1.9 [1.4–2.7] | 72.8 [46.7–86.1] | 0 | 0.164 |
| Married | 4.7 [2.3–9.9] | NA | 1 | 150 | 438 | NA | NA | 1 | NA |
| Occupation, None | 1.9 [0.7–5.1] | NA | 1 | 64 | 64 | NA | NA | 1 | NA |
| Other alcoholic beverages | 2 [0.1–34.2] | NA | 1 | 24 | 48 | NA | NA | 1 | NA |
| Pesticides at home | 9.1 [5.8–14.4] | NA | 1 | 236 | 236 | NA | NA | 1 | NA |
| Presence of HGV | 6.7 [3.1–14.2] | NA | 1 | 167 | 167 | NA | NA | 1 | NA |
| Previous blood transfusion | 7.3 [1.1–48.1] | NA | 2 | 246 | 164 | 3.7 [2–6.8] | 92.6 [75.2–97.8] | 0 | NA |

*(Continued)*

**Table 1.** (*Continued*)

| | OR (95%CI) | 95% Prediction interval | N Studies | N HCC cases | N controls | H (95% CI) | I² (95%CI) | P heterogeneity | P Egger test |
|---|---|---|---|---|---|---|---|---|---|
| Residence in rural area | 6.7 [4.2–10.7] | NA | 2 | 298 | 586 | 1.2 | 29.9 | 0.232 | NA |
| Rodenticides at field | 9.1 [5.8–14.4] | NA | 1 | 236 | 236 | NA | NA | 1 | NA |
| Rodenticides at home | 9.1 [5.8–14.4] | NA | 1 | 236 | 236 | NA | NA | 1 | NA |
| Smoke exposure | 8.3 [5.9–11.6] | [4.7–14.4] | 5 | 697 | 955 | 1 [1–2.1] | 0 [0–77.8] | 0.442 | 0.702 |
| Surgical antecedents | 1.9 [0.7–5.1] | NA | 1 | 64 | 64 | NA | NA | 1 | NA |
| Tattoos-scarifications | 8.8 [2.4–32] | [0–41387963.3] | 3 | 319 | 234 | 2.7 [1.6–4.6] | 86.3 [60.5–95.3] | 0.001 | 0.309 |
| Use of intravenous drug | 1.9 [0.7–5.1] | NA | 1 | 64 | 64 | NA | NA | 1 | NA |
| Aflatoxin B1-albumin adducts (pg_mg) | 2 [0.1–34.2] | NA | 1 | 24 | 48 | NA | NA | 1 | NA |
| Age | 5.5 [3–10.2] | [0.5–58.4] | 4 | 435 | 720 | 1.6 [1–2.8] | 63.1 [0–87.6] | 0.044 | 0.351 |
| Erythrocyte sedimentation rate (mm_hr) | 2 [0.1–34.2] | NA | 1 | 24 | 48 | NA | NA | 1 | NA |
| Hemoglobin (g_dL) | 2 [0.1–34.2] | NA | 1 | 24 | 48 | NA | NA | 1 | NA |
| Transferrin saturation (%) | 2 [0.1–34.2] | NA | 1 | 24 | 48 | NA | NA | 1 | NA |
| **HDV** | | | | | | | | | |
| **Anti-HDV** | | | | | | | | | |
| **Non-hepatic diseases** | | | | | | | | | |
| Overall | 25.8 [5.9–112.3] | [0–353332.5] | 3 | 275 | 150 | 1 [1–2] | 0 [0–74.3] | 0.667 | 0.691 |
| Risk of bias | 30.8 [1.9–509] | NA | 1 | 195 | 49 | NA | NA | 1 | NA |
| Diabetes mellitus | 30.8 [1.9–509] | NA | 1 | 195 | 49 | NA | NA | 1 | NA |
| Diet cassava consumption | 30.8 [1.9–509] | NA | 1 | 195 | 49 | NA | NA | 1 | NA |
| Diet groundnut consumption | 30.8 [1.9–509] | NA | 1 | 195 | 49 | NA | NA | 1 | NA |
| Diet maize consumption | 30.8 [1.9–509] | NA | 1 | 195 | 49 | NA | NA | 1 | NA |
| Diet rice consumption | 30.8 [1.9–509] | NA | 1 | 195 | 49 | NA | NA | 1 | NA |
| Diet sorghum consumption | 30.8 [1.9–509] | NA | 1 | 195 | 49 | NA | NA | 1 | NA |
| Male gender | 25.8 [5.9–112.3] | [0–353332.5] | 3 | 275 | 150 | 1 [1–2] | 0 [0–74.3] | 0.667 | 0.691 |
| Smoke exposure | 30.8 [1.9–509] | NA | 1 | 195 | 49 | NA | NA | 1 | NA |
| Tattoos-scarifications | 8 [0.4–159] | NA | 1 | 51 | 55 | NA | NA | 1 | NA |
| Age | 8 [0.4–159] | NA | 1 | 51 | 55 | NA | NA | 1 | NA |
| **HBV/HCV** | | | | | | | | | |
| **HBsAg/Anti-HCV** | | | | | | | | | |
| **Healthy controls** | | | | | | | | | |
| Overall | 7.8 [4.5–13.7] | [0.2–288.2] | 3 | 588 | 332 | 1 [1–2.7] | 0 [0–86.3] | 0.468 | 0.676 |

(*Continued*)

**Table 1.** (Continued)

| | OR (95%CI) | 95% Prediction interval | N Studies | N HCC cases | N controls | H (95% CI) | I² (95%CI) | P heterogeneity | P Egger test |
|---|---|---|---|---|---|---|---|---|---|
| Risk of bias | 7.9 [4.5–13.9] | NA | 2 | 528 | 302 | 1.2 | 32.9 | 0.222 | NA |
| Cancer family history | 7.4 [4.1–13.1] | NA | 1 | 148 | 150 | NA | NA | 1 | NA |
| Male gender | 7.4 [4.1–13.1] | NA | 1 | 148 | 150 | NA | NA | 1 | NA |
| Married | 7.4 [4.1–13.1] | NA | 1 | 148 | 150 | NA | NA | 1 | NA |
| Smoke exposure | 7.4 [4.1–13.1] | NA | 1 | 148 | 150 | NA | NA | 1 | NA |
| Age | 7.4 [4.1–13.1] | NA | 1 | 148 | 150 | NA | NA | 1 | NA |

apparently healthy controls or those with non-hepatic disease. These results were confirmed when taking into account multiple confounding factors for the risk of developing HCC such as pesticides, aflatoxin B1, alcohol, smoking, other comorbidities, geographic context, mode of diet, gender, and age. Significant associations were seen in all subgroups of HBsAg, HBV DNA, anti-HCV markers with apparently healthy controls and those with non-hepatic disease.

The high risk of HCC associated with HBV, HCV, HDV and HBV/HCV markers observed during this study is consistent with previous systematic reviews conducted at global, regional and national levels [6–10, 66].

The following information should be taken into account when interpreting the findings in this review. The gold standard for diagnosing HCC is histology. In this review of data from Africa, a context where access to HCC diagnosis is restricted [21–23], only half of the included studies had used histology for the diagnosis of HCC. Difficulty in accessing the diagnosis of HCC in the African context may also have overrepresented patients with liver tumors in the last disease stages in this review. More than half of the included studies used multiple diagnostic approaches for HCC, suggesting a substantial amount of residual heterogeneity that we were not able to address in a subgroup analysis. The majority of included study investigators also did not ascertain the absence of HCC in controls, which suggests the possibility of misclassifying cases as controls. Although we planned to include prospective longitudinal studies in this review, all included studies were retrospective case-control studies. This suggests a potential recall bias, which is further exacerbated by the poor prognosis of cases recruited in the terminal phase of HCC. All the studies that we included had a case-control design with a prospective diagnosis of markers of viral hepatitis B, C and D for the majority. It is usually accepted that chronic hepatitis is those that have lasted in patients for at least 6 months. Without the participant follow-up data reported in the included studies, we are unable to distinguish acute from chronic hepatitis in the findings of the present review. Due to insufficient or absence of data [40, 52], for HBV, HCV and HDV, we did not assess the role played by occult infections, viral load, mutations, genotypes, and antiviral therapy on the risk of development of the HCC. Our findings are, however, very robust due to 1) a strengthening of the level of confidence for the overall results by a sensitivity analysis including only comparable studies for the major confounding factors 2) a wide range of markers of past and active HBV, HCV, and HDV infections, and 3) absence of publication bias.

The facies of hepatic viruses circulating in Africa being specific to both genotypes and potential mutations, prospective longitudinal studies are required to determine the role of

these parameters as biomarkers for the early diagnosis of the risk of progression to HCC [18, 19, 67]. Further studies investigating the role of occult viral hepatitis and the role of the viral load and viral therapy in progression to HCC are also required for African populations [11, 68, 69]. In this review with only case-control studies including cases in the terminal phase of the disease with a poor prognosis, large-scale longitudinal studies on healthy populations with prospective follow-up and/or a systematic collection of health data could lead to the identification of more early predictors of progression to HCC, and thus hope to modify the high mortality of HCC cases recorded in Africa. HBV, the predominant etiological agent of HCC, is highly endemic in Africa. Due to the protective effect of HBV treatment on the risk of progression to HCC [70, 71], it would be crucial to improve prevention through strong vaccination policies, diagnostic access and appropriate HBV treatment. Improving access in Africa to new direct-acting HCV therapies could potentially be of importance in reducing the burden of HCC.

Taking into account a wide range of confounders, the findings of this review suggest that HBV/HCV coinfections and HBV, HCV, and HDV infections are associated with a high risk of the occurrence of HCC.

## Supporting information

**S1 Checklist. Preferred reporting items for systematic reviews and meta-analyses checklist.**
(PDF)

**S1 Table. Search strategy in PubMed.**
(PDF)

**S2 Table. Preferred reporting items for systematic reviews and meta-analyses checklist.**
(PDF)

**S3 Table. Items for risk of bias assessment.**
(PDF)

**S4 Table. Main reasons of exclusion of eligible studies.**
(PDF)

**S5 Table. Risk of bias assessment.**
(PDF)

**S6 Table. Characteristics of included studies.**
(PDF)

**S7 Table. Individual characteristics of included studies.**
(PDF)

**S8 Table. P-value of Khi-2 and fisher exact tests for qualitative confounding factors.**
(PDF)

**S9 Table. P-value of student test for quantitative confounding factors.**
(PDF)

**S10 Table. Subgroup analyses of hepatocellular carcinoma development in people with and without viral hepatitis infections in Africa.**
(PDF)

**S1 Fig. Funnel chart for publications of the association between HBeAg in cases and apparently healthy controls and the risk of developing hepatocellular carcinoma.**
(PDF)

**S2 Fig. Funnel chart for publications of the association between HBsAg in cases and apparently healthy controls and the risk of developing hepatocellular carcinoma.**
(PDF)

**S3 Fig. Funnel chart for publications of the association between HBsAg in cases and controls with liver cirrhosis and the risk of developing hepatocellular carcinoma.**
(PDF)

**S4 Fig. Funnel chart for publications of the association between HBsAg in cases and controls with non-hepatic diseases and the risk of developing hepatocellular carcinoma.**
(PDF)

**S5 Fig. Funnel chart for publications of the association between HBsAg in cases and controls with other liver disorders and the risk of developing hepatocellular carcinoma.**
(PDF)

**S6 Fig. Funnel chart for publications of the association between HBV DNA in cases and controls with non-hepatic diseases and the risk of developing hepatocellular carcinoma.**
(PDF)

**S7 Fig. Funnel chart for publications of the association between anti-HCV in cases and apparently healthy controls and the risk of developing hepatocellular carcinoma.**
(PDF)

**S8 Fig. Funnel chart for publications of the association between anti-HCV in cases and controls with liver cirrhosis and the risk of developing hepatocellular carcinoma.**
(PDF)

**S9 Fig. Funnel chart for publications of the association between anti-HCV in cases and controls with non-hepatic diseases and the risk of developing hepatocellular carcinoma.**
(PDF)

**S10 Fig. Funnel chart for publications of the association between anti-HDV in cases and controls with non-hepatic diseases and the risk of developing hepatocellular carcinoma.**
(PDF)

**S11 Fig. Funnel chart for publications of the association between HBV/HCV coinfection in cases and apparently healthy controls and the risk of developing hepatocellular carcinoma.**
(PDF)

## Author Contributions

**Conceptualization:** Donatien Serge Mbaga, Sebastien Kenmoe, Sara Honorine Riwom Essama.

**Data curation:** Donatien Serge Mbaga, Sebastien Kenmoe, Cyprien Kengne-Ndé, Jean Thierry Ebogo-Belobo, Arnol Bowo-Ngandji.

**Formal analysis:** Sebastien Kenmoe, Cyprien Kengne-Ndé.

**Methodology:** Donatien Serge Mbaga, Sebastien Kenmoe, Cyprien Kengne-Ndé, Jean Thierry Ebogo-Belobo, Gadji Mahamat, Joseph Rodrigue Foe-Essomba, Marie Amougou-Atsama, Serges Tchatchouang, Inès Nyebe, Alfloditte Flore Feudjio, Ginette Irma Kame-Ngasse, Jeannette Nina Magoudjou-Pekam, Lorraine K. M. Fokou, Dowbiss Meta-Djomsi, Martin Maïdadi-Foudi, Sabine Aimee Touangnou-Chamda, Audrey Gaelle Daha-Tchoffo, Abdel

Aziz Selly-Ngaloumo, Rachel Audrey Nayang-Mundo, Jacqueline Félicité Yéngué, Jean Bosco Taya-Fokou, Raoul Kenfack-Momo, Efietngab Atembeh Noura, Cynthia Paola Demeni Emoh, Hervé Raoul Tazokong, Arnol Bowo-Ngandji, Carole Stéphanie Sake, Etienne Atenguena Okobalemba, Jacky Njiki Bikoi, Richard Njouom, Sara Honorine Riwom Essama.

**Project administration:** Donatien Serge Mbaga, Sebastien Kenmoe, Richard Njouom, Sara Honorine Riwom Essama.

**Supervision:** Sebastien Kenmoe.

**Validation:** Donatien Serge Mbaga, Sebastien Kenmoe, Cyprien Kengne-Ndé, Jean Thierry Ebogo-Belobo, Gadji Mahamat, Joseph Rodrigue Foe-Essomba, Marie Amougou-Atsama, Serges Tchatchouang, Inès Nyebe, Alfloditte Flore Feudjio, Ginette Irma Kame-Ngasse, Jeannette Nina Magoudjou-Pekam, Lorraine K. M. Fokou, Dowbiss Meta-Djomsi, Martin Maïdadi-Foudi, Sabine Aimee Touangnou-Chamda, Audrey Gaelle Daha-Tchoffo, Abdel Aziz Selly-Ngaloumo, Rachel Audrey Nayang-Mundo, Jacqueline Félicité Yéngué, Jean Bosco Taya-Fokou, Raoul Kenfack-Momo, Efietngab Atembeh Noura, Cynthia Paola Demeni Emoh, Hervé Raoul Tazokong, Arnol Bowo-Ngandji, Carole Stéphanie Sake, Etienne Atenguena Okobalemba, Jacky Njiki Bikoi, Richard Njouom, Sara Honorine Riwom Essama.

**Writing – original draft:** Donatien Serge Mbaga, Sebastien Kenmoe.

**Writing – review & editing:** Donatien Serge Mbaga, Sebastien Kenmoe, Cyprien Kengne-Ndé, Jean Thierry Ebogo-Belobo, Gadji Mahamat, Joseph Rodrigue Foe-Essomba, Marie Amougou-Atsama, Serges Tchatchouang, Inès Nyebe, Alfloditte Flore Feudjio, Ginette Irma Kame-Ngasse, Jeannette Nina Magoudjou-Pekam, Lorraine K. M. Fokou, Dowbiss Meta-Djomsi, Martin Maïdadi-Foudi, Sabine Aimee Touangnou-Chamda, Audrey Gaelle Daha-Tchoffo, Abdel Aziz Selly-Ngaloumo, Rachel Audrey Nayang-Mundo, Jacqueline Félicité Yéngué, Jean Bosco Taya-Fokou, Raoul Kenfack-Momo, Efietngab Atembeh Noura, Cynthia Paola Demeni Emoh, Hervé Raoul Tazokong, Arnol Bowo-Ngandji, Carole Stéphanie Sake, Etienne Atenguena Okobalemba, Jacky Njiki Bikoi, Richard Njouom, Sara Honorine Riwom Essama.

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
