## [Decision Letter · Decision Letter 0]

22 Jun 2021

PONE-D-21-08686

Hepatitis B, C and D virus infections and risk of hepatocellular carcinoma in Africa: A meta-analysis including sensitivity analyses for studies comparable for confounders

PLOS ONE

Dear Dr. MBAGA,

Thank you for submitting your manuscript to PLOS ONE. After careful consideration, we feel that it has merit but does not fully meet PLOS ONE’s publication criteria as it currently stands. Therefore, we invite you to submit a revised version of the manuscript that addresses the points raised during the review process.

The authors need to define chronic hepatitis B, C, and D. Also, the impact of antiviral therapy on HCC development would be studied.

We look forward to receiving your revised manuscript.

Kind regards,

Do Young Kim, MD, PhD

Academic Editor

PLOS ONE

Journal Requirements:

Reviewers' comments:

Reviewer's Responses to Questions

**Comments to the Author**

1. Is the manuscript technically sound, and do the data support the conclusions?

Reviewer #1: Partly

Reviewer #2: Yes

2. Has the statistical analysis been performed appropriately and rigorously? 

Reviewer #1: Yes

Reviewer #2: No

3. Have the authors made all data underlying the findings in their manuscript fully available?

Reviewer #1: Yes

Reviewer #2: Yes

4. Is the manuscript presented in an intelligible fashion and written in standard English?

Reviewer #1: Yes

Reviewer #2: Yes

5. Review Comments to the Author

Reviewer #1: Manuscript ID: PONE-D-21-08686

Manuscript Title: Hepatitis B, C and D virus infections and risk of hepatocellular carcinoma in Africa: A meta-analysis including sensitivity analyses for studies comparable for confounders

This article analyzed the impact of HBV, HCV, or HDV infections on the risk of developing HCC in Africa with a systemic review. This is an interesting research that reflects regional characteristics of Africa. However, there are a couple of points to be addressed in this article.

1. Authors have analyzed the impact of HBV, HCV, or HDV infections on the risk of developing HCC according to the presence of HBeAg, HBsAg, or HBV DNA. However, the presence of chronic HBV, HCV, or HDV infection should be appropriately defined. Authors should re-analyze the OR according to the presence of chronic hepatitis B or chronic hepatitis C. Also, the definition of chronic hepatitis B or C should be described in the Method section.

2. Please show the risk of bias graphs for the analyses.

Reviewer #2: This study investigated the association of viral hepatitis and risk of HCC in African population. Authors performed thorough assessment including other confounders, however, there were several issues to be considered.

1. This meta-analysis found relevant articles through African Index Medicus, and African Journal Online databases, and manual searches. Although I agree with that viral hepatitis could increase the risk of HCC, the reliability of meta-analysis results could be derived from dependable data sources. Therefore, analysis with articles searched from PubMed, MEDLINE, EMBASE, and the Cochrane library would be reasonable for publication.

2. The main finding of this article is not interesting or novel. Also, if meta-analysis regarding HDV infection was impossible, authors should remove HDV from title or manuscript. This article is not a review article.

3. Authors did not present the prevalence of HCC in patients with positive HBeAg, HBsAg, and anti-HCV. Also, the effect of antiviral therapy on HCC was not considered in this study. If authors only included patients without any antiviral therapy, that should be declared as one of inclusion criteria.

4. When collecting characteristics from each article, mean age, proportion of male gender, median time to HCC development should be investigated and presented in table.

6. PLOS authors have the option to publish the peer review history of their article (what does this mean?). If published, this will include your full peer review and any attached files.

Reviewer #1: No

Reviewer #2: No

---

## [Author Response · Author response to Decision Letter 0]

8 Jul 2021

Review Comments to the Author

Editor 

Reviewer #1: 

This article analyzed the impact of HBV, HCV, or HDV infections on the risk of developing HCC in Africa with a systemic review. This is an interesting research that reflects regional characteristics of Africa. However, there are a couple of points to be addressed in this article.

Authors: Thank you for appreciation.

1. Authors have analyzed the impact of HBV, HCV, or HDV infections on the risk of developing HCC according to the presence of HBeAg, HBsAg, or HBV DNA. However, the presence of chronic HBV, HCV, or HDV infection should be appropriately defined. Authors should re-analyze the OR according to the presence of chronic hepatitis B or chronic hepatitis C. Also, the definition of chronic hepatitis B or C should be described in the Method section.

Authors: Thank you for this comment, from the data reported in the included studies, it was not possible for us to pool the studies with participants with chronic hepatitis. We have now reported this additional limit in the discussion as shown below, thank you.

“All the studies that we included had a case-control design with a prospective diagnosis of markers of viral hepatitis B, C and D for the majority. It is usually accepted that chronic hepatitis is those that have lasted in patients for at least 6 months. Without the participant follow-up data reported in the included studies, we are unable to distinguish acute from chronic hepatitis in the findings of the present review.” 

2. Please show the risk of bias graphs for the analyses.

Authors: The S5 Table for Risk of bias assessment in included studies is already provided. The publication bias evaluation by funnel plots are also already provided (S1-11 Figures).

Reviewer #2: This study investigated the association of viral hepatitis and risk of HCC in African population. Authors performed thorough assessment including other confounders, however, there were several issues to be considered.

Authors: Thanks for the comment, we will carefully consider all of your suggestions and do our best to address them all.

1. This meta-analysis found relevant articles through African Index Medicus, and African Journal Online databases, and manual searches. Although I agree with that viral hepatitis could increase the risk of HCC, the reliability of meta-analysis results could be derived from dependable data sources. Therefore, analysis with articles searched from PubMed, MEDLINE, EMBASE, and the Cochrane library would be reasonable for publication.

Authors: As mentioned in the methodology section, in addition to the 2 specific African databases mentioned here, we searched the PubMed and Web of Science databases. We believe that these 4 databases in addition to the manual search that we have done, could guarantee sufficient sensitivity for this review, thank you.

2. The main finding of this article is not interesting or novel. Also, if meta-analysis regarding HDV infection was impossible, authors should remove HDV from title or manuscript. This article is not a review article.

Authors: We totally agree that the number of studies included in the HDV part was very low. We moderated our interpretation to retain only the category that included 3 studies (estimate with anti-HDV and controls with non-hepatic disease). We have modified the text throughout the main manuscript and the figure accordingly, thank you.

3. Authors did not present the prevalence of HCC in patients with positive HBeAg, HBsAg, and anti-HCV. Also, the effect of antiviral therapy on HCC was not considered in this study. If authors only included patients without any antiviral therapy, that should be declared as one of inclusion criteria.

Authors Thank you for these comments. The objective of the review was to compare the prevalence of hepatic viruses B, C and D between patients with and without hepatocellular carcinoma. The prevalence of hepatocellular carcinoma only in patients with HBeAg, HBsAg and anti-HCV positive is outside the scope of our review. We planned to collect data related to all potential sources of heterogeneity and the confounding factors reported by the authors of the included studies, to the best of our knowledge, the included studies did not report if the participants were on antiviral treatment. We have now reported this limit in the discussion section, thank you.

4. When collecting characteristics from each article, mean age, proportion of male gender, median time to HCC development should be investigated and presented in table.

Authors: We dealt in this subject with a comparative approach and we collected the basic data of the patients with and without hepatocellular carcinoma included in the case control studies. In addition to the age and male percentage reported here, we have collected and presented several other baseline data in the S8 and 9 Tables, thank you.

---

## [Decision Letter · Decision Letter 1]

3 Nov 2021

PONE-D-21-08686R1Hepatitis B, C and D virus infections and risk of hepatocellular carcinoma in Africa: A meta-analysis including sensitivity analyses for studies comparable for confoundersPLOS ONE

Dear Dr. MBAGA,

Thank you for submitting your manuscript to PLOS ONE. After careful consideration, we feel that it has merit but does not fully meet PLOS ONE’s publication criteria as it currently stands. Therefore, we invite you to submit a revised version of the manuscript that addresses the points raised during the review process.

 The authors tried to address reviewer's queries. However, the response was not perfect. So, authors need to fully answer the reviewer's comments.

We look forward to receiving your revised manuscript.

Kind regards,

Do Young Kim, MD, PhD

Academic Editor

PLOS ONE

Journal Requirements:

Reviewers' comments:

Reviewer's Responses to Questions

**Comments to the Author**

1. If the authors have adequately addressed your comments raised in a previous round of review and you feel that this manuscript is now acceptable for publication, you may indicate that here to bypass the “Comments to the Author” section, enter your conflict of interest statement in the “Confidential to Editor” section, and submit your "Accept" recommendation.

Reviewer #1: All comments have been addressed

Reviewer #2: (No Response)

2. Is the manuscript technically sound, and do the data support the conclusions?

Reviewer #1: Yes

Reviewer #2: Yes

3. Has the statistical analysis been performed appropriately and rigorously? 

Reviewer #1: Yes

Reviewer #2: No

4. Have the authors made all data underlying the findings in their manuscript fully available?

Reviewer #1: Yes

Reviewer #2: No

5. Is the manuscript presented in an intelligible fashion and written in standard English?

Reviewer #1: Yes

Reviewer #2: No

6. Review Comments to the Author

Reviewer #1: The analyses were performed according to the presence of HBsAg, HBeAg, HBV DNA or HCV RNA. Actually, the odds ratio of developing HCC should be performed according to the presence of chronic hepatitis B or C or the presence of LC. This study is a meta-analysis which systemically reviewed and analyzed the previously published studies, therefore I understand the limitation. Considering the specificity of the region where studies were performed, I think it can be a valuable study.

Reviewer #2: Authors tried to revise their article following the reviewer’s comment, however, still there is unresolved issues.

1. If authors only included patients without any antiviral therapy, that should be declared as one of inclusion criteria.

2. When collecting characteristics from each article, mean age, proportion of male gender, median time to HCC development should be investigated and presented in table. I can’t find those data in S8 or S9 tables.

7. PLOS authors have the option to publish the peer review history of their article (what does this mean?). If published, this will include your full peer review and any attached files.

Reviewer #1: No

Reviewer #2: No

---

## [Author Response · Author response to Decision Letter 1]

6 Nov 2021

Reviewer #1: 

All comments have been addressed.

Authors: Thank you.

Reviewer #1: The analyses were performed according to the presence of HBsAg, HBeAg, HBV DNA or HCV RNA. Actually, the odds ratio of developing HCC should be performed according to the presence of chronic hepatitis B or C or the presence of LC. This study is a meta-analysis which systemically reviewed and analyzed the previously published studies; therefore, I understand the limitation. Considering the specificity of the region where studies were performed, I think it can be a valuable study.

Authors: Thank you.

Reviewer #2: Authors tried to revise their article following the reviewer’s comment, however, still there is unresolved issues.

1. If authors only included patients without any antiviral therapy, that should be declared as one of inclusion criteria.

Authors: Thanks to the Reviewer for this comment. We understand that antiviral therapy for hepatitis B, C or D may be a major factor influencing the development of hepatocellular carcinoma. We systematically collected from the included studies and presented all reported socio-demographic and clinical, quantitative and qualitative variables likely to influence the association between viral hepatitis (B, C and D) and hepatocellular carcinoma in Africa. We have carefully reviewed all of the included studies and to the best of our knowledge, the information on antiviral treatment for hepatitis is unclear and/or reported. We would have performed a subgroup analysis for studies with patients on and without viral therapy to highlight this factor as a potential source of heterogeneity in our estimates. Unfortunately, these data were not found in the included studies. We have now clarified in the main manuscript that our inclusion criteria did not depend on the antiviral therapy status to remove any potential confusion.

2. When collecting characteristics from each article, mean age, proportion of male gender, median time to HCC development should be investigated and presented in table. I can’t find those data in S8 or S9 tables.

Authors: For each included study, we systematically collected the reported mean age and standard deviation for cases and controls. This information is presented in the S9 Table. In line 2 of the S9 table, for example, for the study by Amr et al., 2014, the mean age reported is 52.2 years for cases and 45.4 years for controls. We also systematically collected the number of male subjects for cases and controls. This information is presented in the S8 Table. In line 5 of the S8 Table, for example, for the study by Bahri et al., 2011, the number of reported males is 100 for cases and 152 for controls.

For the median time to development of HCC, this information is not clear and/or reported in the included studies. This could also be understood from the fact that all the included studies are case controls which are characterized by retrospective exposure (viral hepatitis infection) of which the exact time of occurrence is not known.

---

## [Decision Letter · Decision Letter 2]

10 Jan 2022

Hepatitis B, C and D virus infections and risk of hepatocellular carcinoma in Africa: A meta-analysis including sensitivity analyses for studies comparable for confounders

PONE-D-21-08686R2

Dear Dr. MBAGA,

We’re pleased to inform you that your manuscript has been judged scientifically suitable for publication and will be formally accepted for publication once it meets all outstanding technical requirements.

Kind regards,

Do Young Kim, MD, PhD

Academic Editor

PLOS ONE

Additional Editor Comments (optional):

Reviewers' comments:

Reviewer's Responses to Questions

**Comments to the Author**

1. If the authors have adequately addressed your comments raised in a previous round of review and you feel that this manuscript is now acceptable for publication, you may indicate that here to bypass the “Comments to the Author” section, enter your conflict of interest statement in the “Confidential to Editor” section, and submit your "Accept" recommendation.

Reviewer #1: All comments have been addressed

Reviewer #2: All comments have been addressed

2. Is the manuscript technically sound, and do the data support the conclusions?

Reviewer #1: Yes

Reviewer #2: Yes

3. Has the statistical analysis been performed appropriately and rigorously? 

Reviewer #1: Yes

Reviewer #2: Yes

4. Have the authors made all data underlying the findings in their manuscript fully available?

Reviewer #1: Yes

Reviewer #2: Yes

5. Is the manuscript presented in an intelligible fashion and written in standard English?

Reviewer #1: Yes

Reviewer #2: Yes

6. Review Comments to the Author

Reviewer #1: (No Response)

Reviewer #2: All comments have been addressed with your best effort.

Although there are some uncertainties in this data, it seems to have some important clinical implications in specific area.

7. PLOS authors have the option to publish the peer review history of their article (what does this mean?). If published, this will include your full peer review and any attached files.

Reviewer #1: No

Reviewer #2: **Yes: **Lee Han Ah

---

## [Editor Report · Acceptance letter]

12 Jan 2022

PONE-D-21-08686R2 

Hepatitis B, C and D virus infections and risk of hepatocellular carcinoma in Africa: A meta-analysis including sensitivity analyses for studies comparable for confounders. 

Dear Dr. Mbaga:

I'm pleased to inform you that your manuscript has been deemed suitable for publication in PLOS ONE. Congratulations! Your manuscript is now with our production department. 

Kind regards, 

on behalf of

Prof. Do Young Kim 

Academic Editor

PLOS ONE